# Myelination and excitation-inhibition balance synergistically shape structure-function coupling across the human cortex

Panagiotis Fotiadis [1,2] ✉, Matthew Cieslak [3], Xiaosong He [4], Lorenzo Caciagli[2], Mathieu Ouellet [2], Theodore D. Satterthwaite [3], Russell T. Shinohara[5,6] & Dani S. Bassett [2,3,7,8,9,10] ✉

Recent work has demonstrated that the relationship between structural and functional connectivity varies regionally across the human brain, with reduced coupling emerging along the sensory-association cortical hierarchy. The biological underpinnings driving this expression, however, remain largely unknown. Here, we postulate that intracortical myelination and excitation-inhibition (EI) balance mediate the heterogeneous expression of structure-function coupling (SFC) and its temporal variance across the cortical hierarchy. We employ atlas- and voxel-based connectivity approaches to analyze neuroimaging data acquired from two groups of healthy participants. Our findings are consistent across six complementary processing pipelines: 1) SFC and its temporal variance respectively decrease and increase across the unimodal-transmodal and granular-agranular gradients; 2) increased myelination and lower EI-ratio are associated with more rigid SFC and restricted moment-to-moment SFC fluctuations; 3) a gradual shift from EI-ratio to myelination as the principal predictor of SFC occurs when traversing from granular to agranular cortical regions. Collectively, our work delivers a framework to conceptualize structure-function relationships in the human brain, paving the way for an improved understanding of how demyelination and/or EI-imbalances induce reorganization in brain disorders.

The structural and functional connectivity patterns of the human brain have been extensively mapped using macroscale neuroimaging. To elucidate how the anatomical wiring of the brain sculpts its functional connectivity in support of flexible cognition, recent studies have increasingly focused on the extent to which structure and function are coupled across brain regions[1–3]. A brain region's 'structure-function coupling' (SFC) refers to the manner in which its functional and structural connectivity statistically depend upon one another. Here, a structural connection is the white matter projections linking two brain regions, as measured by diffusion magnetic resonance imaging (MRI),

[1]Department of Neuroscience, Perelman School of Medicine, University of Pennsylvania, Philadelphia, PA 19104, USA. [2]Department of Bioengineering, University of Pennsylvania, Philadelphia, PA 19104, USA. [3]Department of Psychiatry, Perelman School of Medicine, University of Pennsylvania, Philadelphia, PA 19104, USA. [4]Department of Psychology, University of Science and Technology of China, Hefei, Anhui 230026, China. [5]Penn Statistics in Imaging and Visualization Center, Department of Biostatistics, Epidemiology, and Informatics, University of Pennsylvania, Philadelphia, PA 19104, USA. [6]Center for Biomedical Image Computing & Analytics, University of Pennsylvania, Philadelphia, PA 19104, USA. [7]Department of Electrical & Systems Engineering, University of Pennsylvania, Philadelphia, PA 19104, USA. [8]Department of Physics & Astronomy, University of Pennsylvania, Philadelphia, PA 19104, USA. [9]Department of Neurology, Perelman School of Medicine, University of Pennsylvania, Philadelphia, PA 19104, USA. [10]Santa Fe Institute, Santa Fe, NM 87501, USA. ✉e-mail: panosf@pennmedicine.upenn.edu; dsb@seas.upenn.edu

whereas a functional connection is the statistical similarity between hemodynamic responses arising from two brain regions, as measured by functional MRI (fMRI). Intuitively, a brain region with high SFC has a stronger statistical correlation between its structural and functional connectivity to other regions in the brain.

Regional variations in SFC among individuals track differences in cognitive performance. For example, enhanced working memory performance is correlated with weaker SFC in the unimodal somatosensory cortex and with stronger coupling in transmodal regions within the fronto-parietal and default mode networks[4]. Further, individual differences in SFC predict cognitive flexibility in a perceptual switching task[5], as well as composite cognition scores encompassing multiple cognitive domains[6]. Beyond tracking individual differences in cognition, SFC is altered in a range of neurological and psychiatric disorders, including mild cognitive impairment and Alzheimer's disease[7-9], stroke[10,11], Parkinson's disease[12,13], multiple sclerosis[14,15], epilepsy[16,17], bipolar disorder[18,19], and schizophrenia[20].

In parallel, multiple lines of evidence from studies in healthy individuals have consistently demonstrated that the macroscale coupling of structure and function varies spatially, with a gradual reduction in coupling emerging along a cognitive representational hierarchy[1,4,21-23]. Specifically, evolutionarily conserved primary sensory (unimodal) regions such as visual and somatomotor cortices display relatively strong SFC, whereas evolutionarily rapidly-expanded transmodal association regions such as limbic and default mode areas display weaker SFC[6,21-23]. The presence of a dynamic SFC landscape along the sensory-association hierarchy is thought to foster the emergence of a wide range of functional responses untethered from the underlying anatomical backbone, in turn supporting flexible cognition[4,5,24-26].

Understanding precisely *why* the coupling between structure and function varies across different brain regions is a key challenge in the field[4,21,22]. Insight could be gained by examining how SFC varies across different—yet complementary—types of cortical hierarchies defined by cyto-architectonic and functional properties, as such an examination could clarify to what extent SFC captures the brain's microscale cyto-architectonic and macroscale functional principles. Complementary insights could also be gained by pinpointing specific biological substrates that statistically track (and conceptually explain) regional variation in SFC.

Recent evidence suggests that the differential expression of neuronal circuit properties—including intracortical myelination and synaptic excitation or inhibition—could serve as such biological substrates. Histological and neuroimaging studies show that high-SFC areas in the primary sensory and motor cortex are heavily myelinated, whereas lower-SFC areas in the association cortex are less myelinated[6,25,27-31]. Moreover, regional heterogeneities in intracortical myelination have been linked to differences in functional connectivity patterns across the cortical mantle; brain regions with similar intracortical myelin profiles typically display stronger functional connectivity to each other[29,32]. This correspondence is particularly high within unimodal brain regions; transmodal regions such as the posteromedial cortex, the anterior insular cortex, and the superior portions of the inferior parietal lobule, instead, display a lower correspondence between intracortical myelination and functional connectivity, even after correcting for inter-regional proximity[29]. Lastly, the relationship between structural and functional connectivity drastically changes throughout normative development—a critical period of enhanced neuroplasticity and myelination—which could point towards intracortical myelination's potential involvement as one of its mediators[4,33].

Besides intracortical myelination, neuromodulation has also been implicated as a potential driving factor determining to what extent the brain's functional expression is tethered to the underlying anatomical connectivity. Following a similar spatial pattern as intracortical myelination, synaptic excitation increases from unimodal sensory to transmodal association cortex, tracking a concomitant increase in dendritic complexity and spine count[34]. Further, immunostaining investigations tracking the differential expression of inhibitory neuron subtypes, evince a unimodal-transmodal gradient of dynamic inhibitory control[34,35]. Put together, the ratio between excitatory and inhibitory receptor densities (EI-ratio) appears to increase along the sensory-association hierarchy[36]. What is more, recent work looking into the differences in SFC between patients with Parkinson's disease and healthy controls identified an increased association between the expression of various neurotransmitter receptor genes and disease-related structure-function decoupling[12]. Thus, given that such neuromodulatory systems typically alter the balance between the excitation and inhibition of their targeted neuronal circuits, we postulated that EI-ratio would also play an important role in shaping the healthy human brain's SFC.

The aforementioned observations collectively motivated our hypothesis that the differential expression of intracortical myelination and EI-ratio formally mediate the heterogeneous expression of SFC across the cortex. Here, we use neuroimaging data acquired from two groups of healthy participants and analyzed using six image processing pipelines, to address three complementary aims (Fig. 1). First, to determine *whether* SFC captures macroscale functional and microscale cyto-architectonic principles, we assess the spatial distribution of SFC along four cortical gradients spanning the unimodal (sensory)-transmodal (association) hierarchy: two functional gradients and two cyto-architectonic gradients. Second, to determine *why* SFC varies across the brain, we examine the relationship between SFC and two biological substrates of interest—intracortical myelination and EI-ratio. Third and finally, by combining elements from the two previous aims, we investigate *how* SFC is dynamically shaped by these biological substrates across different cyto-architectonic systems of varying laminar differentiation. Collectively, this work aims to elucidate the biological factors that explain the heterogeneous coupling between structural and functional connectivity across the human cortex.

## Results
### Structure-function coupling variations along the cortical hierarchy
We first examined the heterogeneous expression of SFC and its temporal variance (Methods: Structure-Function Coupling) across the unimodal (sensory)-transmodal (association) hierarchy in 100 unrelated subjects drawn from the Human Connectome Project (HCP) (Methods: Datasets). For this purpose, the unimodal-transmodal hierarchy was characterized using four complementary cortical annotations (Methods: Cortical Hierarchies): two derived from annotating the cortex according to macroscale functional connectivity profiles (the coarse 7 resting-state systems[37] and the continuous principal functional gradient[38]), and two derived from annotating the cortex according to microscale cyto-architectonic profile similarities (the coarse 5 von Economo/Koskinas-inspired cyto-architectonic classes[39] and the continuous "BigBrain" gradient[40,41]). These four annotations were chosen to broadly canvas the space of sensory-association hierarchy from the lenses of both macroscale functional and microscale cyto-architectonic organization.

After parcellating each HCP subject's cortex into spatially contiguous regions (Schaefer parcellation[42]; 400 brain regions), we computed each brain region's average SFC across subjects and designated its regional membership into each of the four aforementioned cortical annotations. In the 7 resting-state systems, SFC was highest in the primary visual and somatomotor cortices, intermediate in the default mode, dorsal attention, fronto-parietal, and ventral attention association systems, and lowest in the limbic system (Fig. 2A; Supplemental Table 1). A decrease in SFC along the unimodal-transmodal hierarchy was also evident along the principal functional gradient, in the form of a significant negative correlation between a brain region's SFC and its

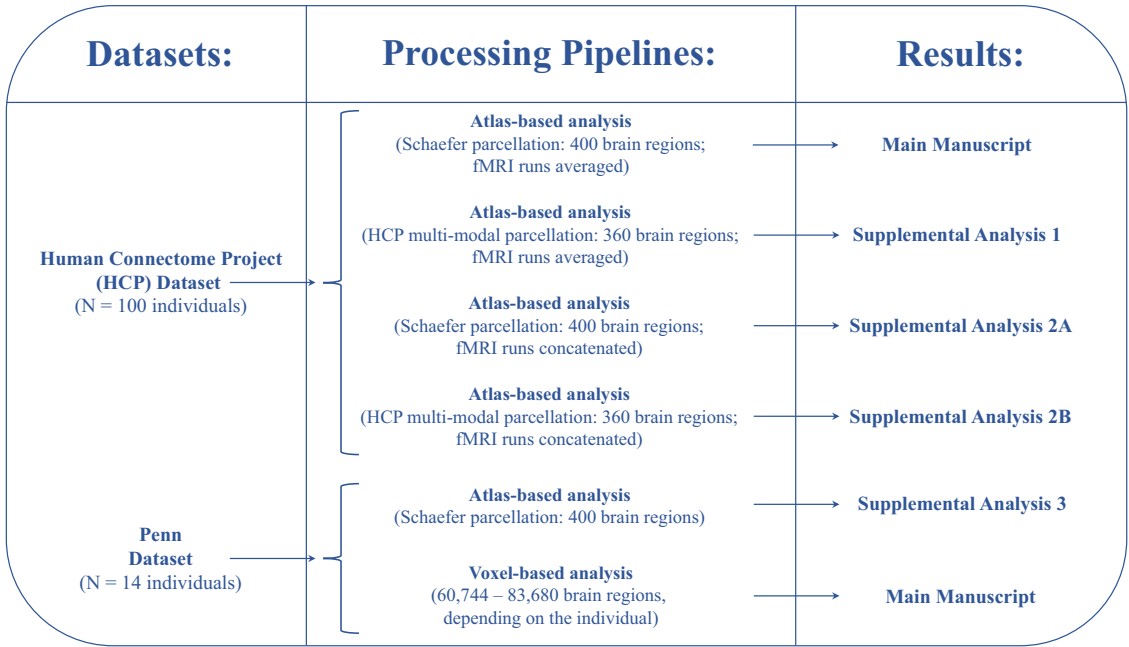

**Fig. 1 | Methodological pipeline.** Schematic illustration of the processing pipelines used to analyze our datasets. The first column corresponds to the datasets used in this study, the second column to the processing pipelines used to analyze each dataset, and the third column to the location of the results of each analysis.

assigned principal gradient scalar (Fig. 2C; $r = -0.35$; $p_{spin} = 0.010$); lower assignments within this gradient capture primary sensory and motor regions, whereas higher assignments capture regions within the default mode network. Across the 5 cyto-architectonic classes, SFC gradually decreased from granular (typically capturing sensory regions)[30,43,44] to agranular (typically capturing motor and association regions)[30,43,44] types and displayed its lowest value in the polar cortical type (Fig. 2B; Supplemental Table 2). Similarly, we observed a significant negative correlation between a brain region's SFC and its assigned location along the BigBrain gradient of microstructure profile covariance (Fig. 2D; $r = -0.42$; $p_{spin} = 0.020$); primary sensory regions occupy the lower end of this gradient while limbic regions represent its apex.

Next, in order to examine how much SFC deviated from its mean value over time, we assessed its moment-to-moment variance throughout the duration of the resting-state fMRI scan. Specifically, we computed each brain region's average temporal SFC variance across subjects (Methods: Processing Pipelines: Functional Connectivity; Supplemental Material: Eq. 1) and examined its heterogeneous expression along the unimodal (sensory)-transmodal (association) hierarchy. In contrast to SFC, temporal SFC variance was highest in the limbic system, intermediate in the default mode, fronto-parietal, dorsal attention, and ventral attention systems, and lowest in the primary visual and somatomotor cortices (Fig. 3A; Supplemental Table 3); a significant increase in temporal SFC variance was observed along the unimodal-transmodal hierarchy, as captured by the principal gradient (Fig. 3C; $r = 0.43$; $p_{spin} = 3.5 \times 10^{-4}$). Using cyto-architectonic annotations, temporal SFC variance (unlike SFC itself) was highest in the polar cortical type; the remaining 4 cortical types displayed—for the most part—similar degrees of temporal SFC variance (Fig. 3B; Supplemental Table 4). Under the more continuous BigBrain gradient, we observed a significant positive correlation between a brain region's temporal SFC variance and its assigned location along the gradient (Fig. 3D; $r = 0.43$; $p_{spin} = 0.003$). To ensure that the correlations observed between a brain region's temporal SFC variance and its location in the sensory-association hierarchy (as shown in Fig. 3C and 3D) were not confounded by the presence of any outlier regions, we repeated the aforementioned analyses after excluding the outlier brain regions. An

outlier brain region was defined as one that exhibited a temporal SFC variance at least three standard deviations away from the mean ($n = 7$). Consistent with our results when the outliers were included, temporal SFC variance was significantly correlated with both the principal functional gradient ($r = 0.42$; $p_{spin} = 4 \times 10^{-4}$) and the BigBrain gradient ($r = 0.40$; $p_{spin} = 0.005$).

To evaluate the reproducibility of our findings, we repeated the above analyses (i) using a different widely-used cortical parcellation (HCP multi-modal parcellation[45]; Supplemental Material: Supplemental Analysis 1), (ii) using a complementary definition of functional signal time series (see Methods: Processing Pipelines: Functional Connectivity; Supplemental Material: Supplemental Analysis 2), and (iii) on the complementary Penn sample to establish generalizability across different subject samples (Supplemental Material: Supplemental Analysis 3). We observed consistent results across all supplemental analyses.

In order to further investigate whether our findings were influenced by the spatial scale of the cortical parcels used (400 and 360 brain regions, respectively), we repeated the above analyses using an independent sample of healthy adults scanned at the University of Pennsylvania ($n = 14$) with particularly high-resolution diffusion spectrum imaging (DSI). Capitalizing on this sample's higher-resolution diffusion scans than the HCP sample's, the data were processed at the voxel level such that each subject's cortical voxel was designated as a separate region (number of regions ranged between 60,744 and 83,680, depending on the subject; Methods: Voxel-based approach).

As above, for each subject we computed each cortical voxel's SFC and determined its membership into the four cortical annotations. Similar to the atlas-based results, we observed a decrease in SFC along the unimodal (sensory)-transmodal (association) hierarchy. The primary somatomotor and limbic cortices displayed the highest and lowest SFC, respectively, within the 7 resting-state systems (Supplemental Fig. 13A; Supplemental Table 25). Further, we observed a significant negative association between SFC and the assigned principal functional gradient scalar across subjects (Supplemental Fig. 13C; mean $r = -0.16$; range: [−0.24, −0.05]; $p_{fisher} < 10^{-4}$). Within the 5 cyto-architectonic types, SFC gradually decreased from granular to agranular types and displayed its lowest value in the polar type (Supplemental Fig. 13B; Supplemental Table 26).

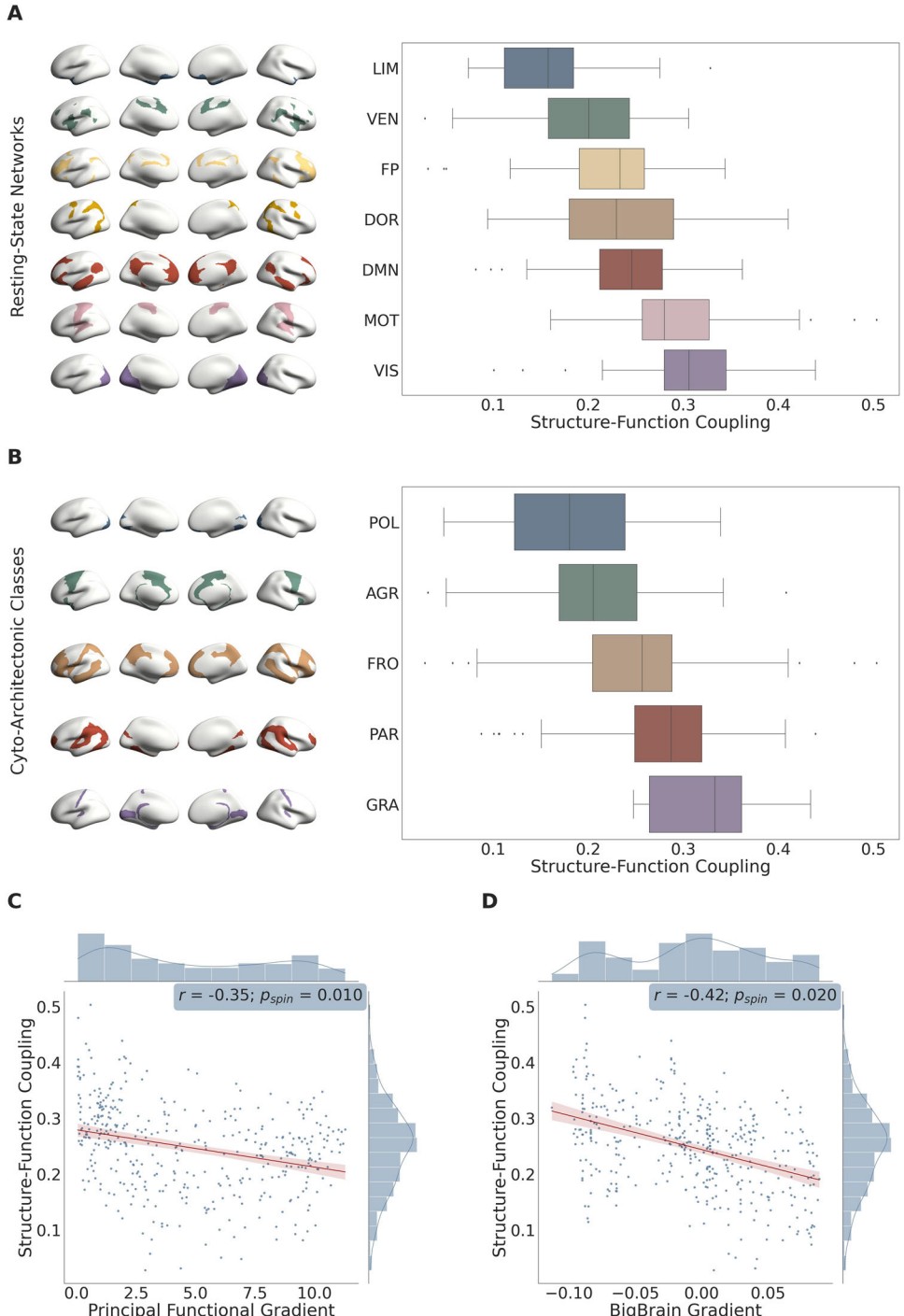

**Fig. 2 | Regional variations in structure-function coupling: atlas-based analysis.** **A** Mean differences in structure-function coupling across the 7 resting-state functional systems (generated using the 100 unrelated HCP subjects and Schaefer 400 atlas; $n = 400$ brain regions/datapoints). Data are presented as boxplots (median value at center line, lower quartile at left bound, upper quartile at right bound) with whiskers extending towards the minimum and maximum non-outlier values of the data; single datapoints denote outliers. The brain regions in each functional system are overlayed on the standardized *fsaverage* brain's surface and illustrated on the left side. LIM Limbic, VEN Ventral Attention, FP Fronto-Parietal, DMN Default Mode Network, DOR Dorsal Attention, MOT Somatomotor, VIS Visual. **B** Mean differences in structure-function coupling across the 5 cyto-architectonic classes (generated using the 100 unrelated HCP subjects and Schaefer 400 atlas; $n = 400$ brain regions/datapoints). Data are presented as boxplots (median value at center line, lower quartile at left bound, upper quartile at right bound) with whiskers extending towards the minimum and maximum non-outlier values of the data; single

datapoints denote outliers. The brain regions involved within each class are overlayed on the standardized *fsaverage* brain's surface and illustrated on the left side. POL Polar, AGR Agranular, FRO Frontal, PAR Parietal, GRA Granular. **C** Scatterplot between the principal functional gradient scalar of each brain region and its corresponding structure-function coupling ($n = 400$ brain regions/datapoints). A linear regression was fit along with a 95% confidence interval (shown in red); the correlation coefficient (two-tailed Spearman's $\rho$: $r$), $p$-value corresponding to the spatial permutation test ($p_{spin}$), and histograms corresponding to each variable are reported. **D** Scatterplot between the "BigBrain" gradient scalar of each brain region and its corresponding structure-function coupling ($n = 400$ brain regions/datapoints). A linear regression was fit along with a 95% confidence interval (shown in red); the correlation coefficient (two-tailed Spearman's $\rho$: $r$), $p$-value corresponding to the spatial permutation test ($p_{spin}$), and histograms corresponding to each variable are reported. Source data are provided as a Source Data file.

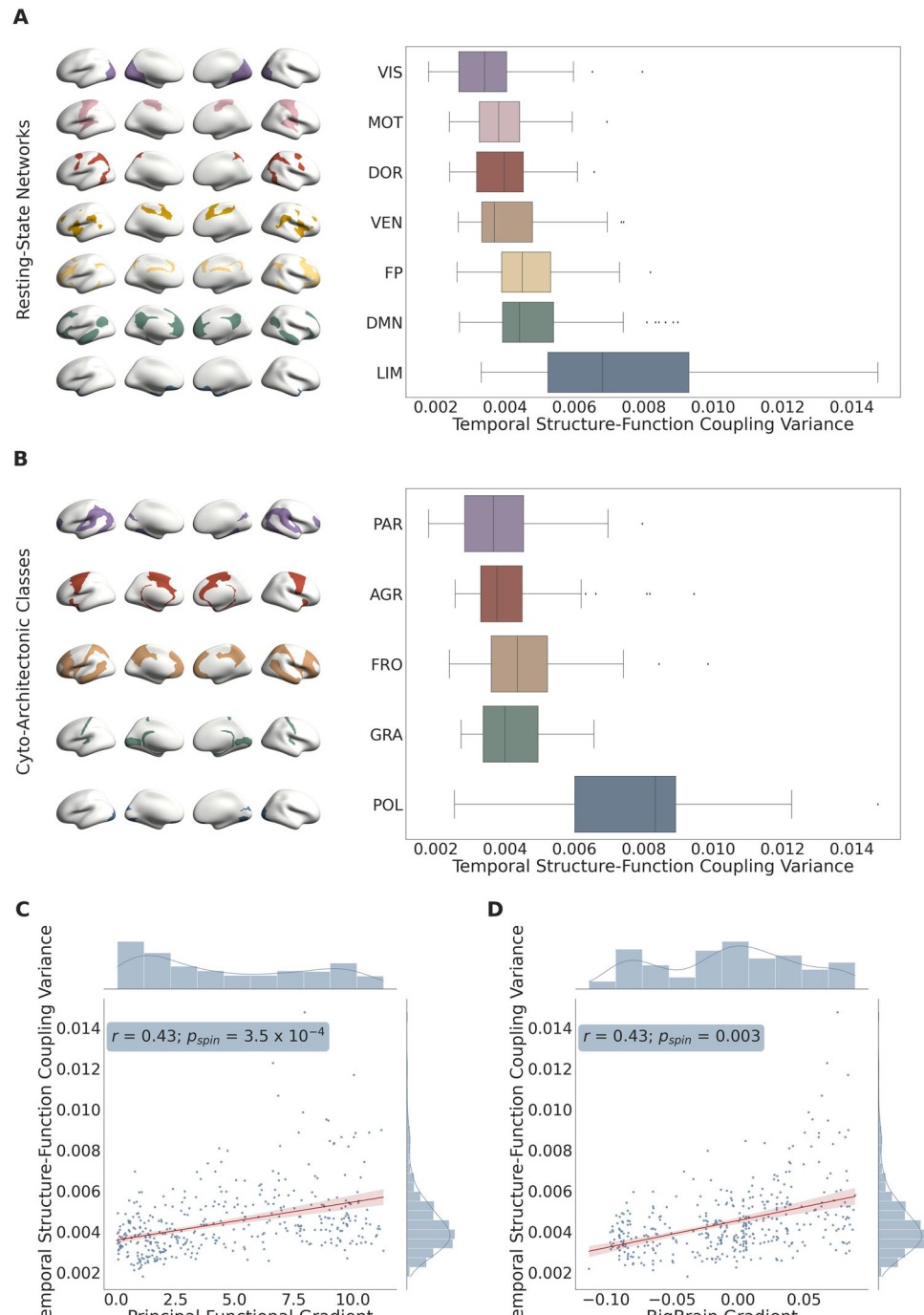

**Fig. 3 | Regional variations in temporal structure-function coupling variance: atlas-based analysis. A** Mean differences in temporal structure-function coupling variance across the 7 resting-state functional systems (generated using the 100 unrelated HCP subjects and Schaefer 400 atlas; $n = 400$ brain regions/datapoints). Data are presented as boxplots (median value at center line, lower quartile at left bound, upper quartile at right bound) with whiskers extending towards the minimum and maximum non-outlier values of the data; single datapoints denote outliers. The brain regions involved within each functional system are overlayed on the standardized *fsaverage* brain's surface and illustrated on the left side. DOR Dorsal Attention, VIS Visual, MOT Somatomotor, VEN Ventral Attention, FP Fronto-Parietal, DMN Default Mode Network, LIM Limbic. **B** Mean differences in temporal structure-function coupling variance across the 5 cyto-architectonic classes (generated using the 100 unrelated HCP subjects and Schaefer 400 atlas; $n = 400$ brain regions/datapoints). Data are presented as boxplots (median value at center line, lower quartile at left bound, upper quartile at right bound) with whiskers extending towards the minimum and maximum non-outlier values of the data; single

datapoints denote outliers. The brain regions involved within each class are overlayed on the standardized *fsaverage* brain's surface and illustrated on the left side. PAR Parietal, AGR Agranular, FRO Frontal, GRA Granular, POL Polar. **C** Scatterplot between the principal functional gradient scalar of each brain region and its corresponding temporal structure-function coupling variance ($n = 400$ brain regions/datapoints). A linear regression was fit along with a 95% confidence interval (shown in red); the correlation coefficient (two-tailed Spearman's $\rho$: $r$), $p$-value corresponding to the spatial permutation test ($p_{spin}$), and histograms corresponding to each variable are reported. **D** Scatterplot between the "BigBrain" gradient scalar of each brain region and its corresponding temporal structure-function coupling variance ($n = 400$ brain regions/datapoints). A linear regression was fit along with a 95% confidence interval (shown in red); the correlation coefficient (two-tailed Spearman's $\rho$: $r$), $p$-value corresponding to the spatial permutation test ($p_{spin}$), and histograms corresponding to each variable are reported. Source data are provided as a Source Data file.

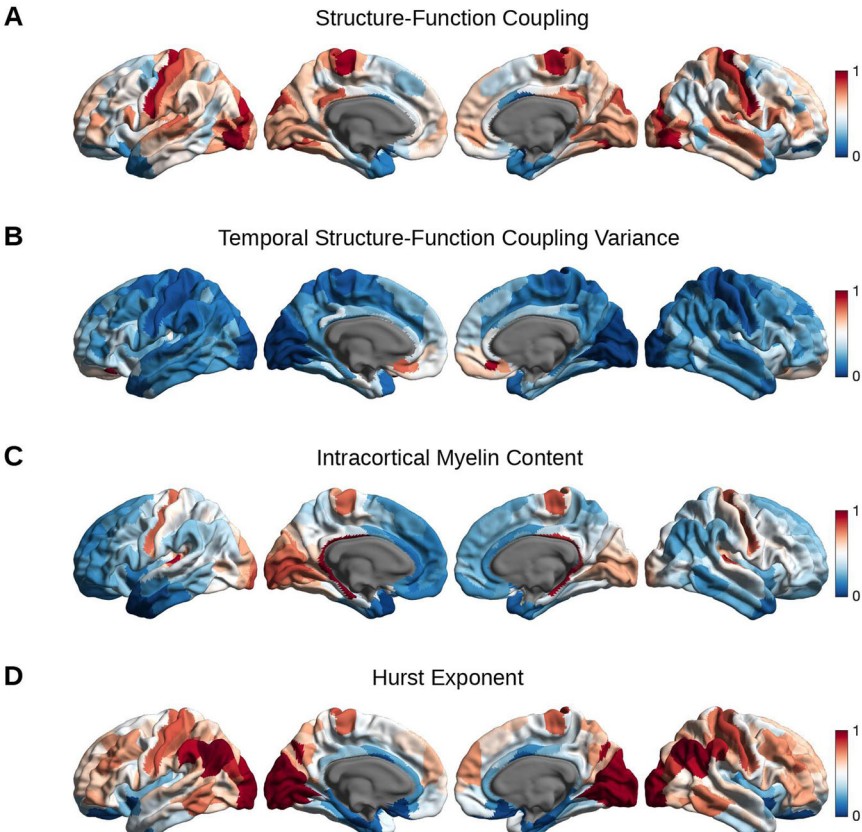

**Fig. 4 | Spatial distributions of the variables of interest.** Schematic of the spatial cortical maps corresponding to structure-function coupling (**A**), temporal structure-function coupling variance (**B**), T1-weighted/T2-weighted signal intensity ratio as a proxy of intracortical myelin content (**C**), and the Hurst exponent of the functional signal time series as a proxy of excitation-inhibition balance (**D**). For visualization purposes, each variable of interest was normalized between 0 and 1 and mapped onto the standardized *Conte69* surface space; the medial wall was excluded from the schematic and is shown in dark gray.

We next computed each cortical voxel's temporal SFC variance across subjects. In general agreement with the atlas-based results, the transmodal default mode and limbic systems displayed the highest temporal SFC variance. The dorsal and ventral attention systems displayed the lowest temporal SFC variance (Supplemental Fig. 14A; Supplemental Table 27). Along the principal functional gradient, there was a prominent increase in temporal SFC variance as one traversed from lower to higher assigned gradient scalars (Supplemental Figure 14C; mean $r = 0.16$; range: [0.01, 0.35]; $p_{fisher} < 10^{-4}$). As for the 5 cyto-architectonic classes, temporal SFC variance was highest in the frontal type and lowest in the agranular type (Supplemental Fig. 14B; Supplemental Table 28).

### Biological correlates of structure-function coupling: whole-brain perspective

To better understand *why* SFC and temporal SFC variance vary across the unimodal (sensory)-transmodal (association) hierarchy, we next examined their relation to two microstructural markers: intracortical myelination and EI-ratio (Fig. 4). Both markers were assessed by non-invasive neuroimaging using previously established approaches. Intracortical myelination was estimated using the subjects' T1-weighted/T2-weighted ratio signal intensity, whereby a greater intensity reflects greater intracortical myelination (Methods: Intracortical Myelination)[28]. The EI-ratio was quantified using the functional signal time series' Hurst exponent, whereby a smaller exponent reflects a heightened EI-ratio (Methods: Excitation-Inhibition Balance)[46].

Across the 400 brain regions defined by the Schaefer parcellation, we observed a significant positive correlation between SFC and intracortical myelin content (Fig. 5A; $r = 0.49$; $p_{spin} = 1.5 \times 10^{-4}$), and a negative correlation between temporal SFC variance and intracortical myelin content (Fig. 5B; $r = -0.29$; $p_{spin} = 0.015$). Higher SFC values corresponded to larger Hurst exponents and thus a decreased EI-ratio (Fig. 5C; $r = 0.46$; $p_{spin} < 10^{-4}$), whereas higher temporal variance in SFC corresponded to lower Hurst exponents and thus a heightened EI-ratio (Fig. 5D; $r = -0.38$; $p_{spin} = 0.004$). Notably, there was no significant association between SFC and the Hurst exponent across the different temporal windows (average Spearman's ρ across brain regions: 0.04; $p_{fisher\ (FDR-corrected)} = 1$), indicating that SFC and EI-ratio do not co-fluctuate over short periods of time (i.e., the duration of the fMRI scan) when examined on the macroscale level.

To ensure that the association between a region's SFC and either biological marker was independent of the other marker and also independent from that region's position along the cortical hierarchy, we re-examined the above relationships using multiple linear regression models. We found that SFC (dependent variable) was independently and positively correlated with intracortical myelin content ($\beta_{stand} = 0.359$; 95% non-parametric bootstrap confidence interval [$BCI$] = [0.357, 0.360]; $p < 10^{-4}$; variance inflation factor [$VIF$] = 1.84) and with the Hurst exponent ($\beta_{stand} = 0.420$; 95% $BCI$ = [0.421, 0.423]; $p < 10^{-4}$; $VIF = 1.25$), after adjusting for the other biological marker, the interaction effect between intracortical myelination and the Hurst exponent, as well as the principal gradient and BigBrain scalar assignments. Further, the correspondence between temporal SFC variance (dependent variable) and the Hurst exponent ($\beta_{stand} = -0.378$; 95% $BCI$ = [−0.379, −0.377]; $p < 10^{-4}$; $VIF = 1.25$), but not intracortical myelin content ($\beta_{stand} = 0.077$; 95% $BCI$ = [0.075, 0.077]; $p = 0.12$; $VIF = 1.84$), remained significant after adjusting for the other marker, the interaction effect between intracortical myelination and the Hurst

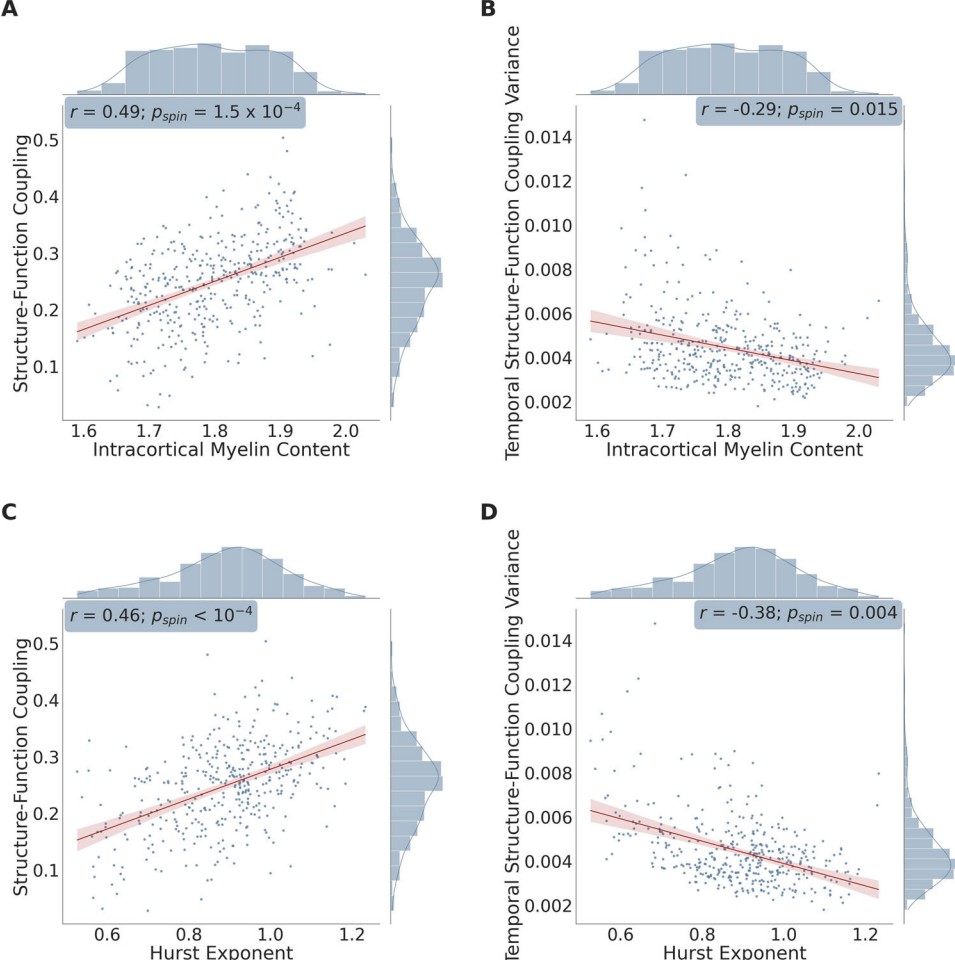

**Fig. 5 | Scatterplots between the variables of interest: atlas-based analysis.**
Scatterplot showing the association between each brain region's: mean structure-function coupling and intracortical myelin content as estimated by the T1-weighted/T2-weighted signal intensity ratio (**A**), mean temporal structure-function coupling variance and intracortical myelin content (**B**), mean structure-function coupling and the Hurst exponent of the functional signal time series (**C**), and mean temporal structure-function coupling variance and the Hurst exponent of the functional signal time series (**D**). For each scatterplot, a linear regression was fit along with a 95% confidence interval (shown in red); correlation coefficients (two-tailed Spearman's $\rho$: $r$), $p$-values corresponding to the spatial permutation test ($p_{spin}$), and histograms corresponding to each variable are displayed. $n = 400$ brain regions in all panels. Source data are provided as a Source Data file.

exponent, and the principal gradient and BigBrain scalar assignments. Notably, the interaction effect between intracortical myelination and the Hurst exponent was significant within this model ($\beta_{stand} = 0.206$; 95% $BCI = [0.205, 0.206]$; $p < 10^{-4}$; $VIF = 1.08$); thus, a potential causal relationship between temporal SFC variance, intracortical myelination, and the Hurst exponent was further explored via a mediation model. Notably, the Hurst exponent was found to significantly mediate the correlation between intracortical myelination and temporal SFC variance (total effect = $-0.0058$; $p < 10^{-4}$, indirect effect = $-0.0014$; BCI = $[-0.0024, -0.0006]$). In other words—and according to this mediation model—the Hurst exponent (i.e., EI-ratio) accounted for 24.1% of the correlation between intracortical myelination and temporal SFC variance.

To assess reproducibility and robustness across different processing pipelines and subject samples, we repeated all aforementioned analyses (i) using the HCP multi-modal cortical parcellation, (ii) using a complementary definition of functional signal time series (see Methods: Processing Pipelines: Functional Connectivity), and (iii) across the Penn sample, and observed consistent results (Supplemental Analyses 1, 2, 3).

To complement our atlas-based results, we also evaluated the relationships between SFC, temporal SFC variance, intracortical myelination, and the Hurst exponent at the voxel level. Across the

cortical voxels, there was once again a positive correlation between SFC and intracortical myelin content (Fig. 6A; Supplemental Fig. 15A; mean $r = 0.11$; range: $[0.08, 0.18]$; $p_{fisher} < 10^{-4}$), and a negative correlation between temporal SFC variance and intracortical myelin content (Fig. 6B; Supplemental Fig. 15B; mean $r = -0.06$; range: $[-0.13, -0.01]$; $p_{fisher} < 10^{-4}$). Stronger SFC was also associated with decreased EI-ratio in the form of higher Hurst exponents (Fig. 6C; Supplemental Figure 15C; mean $r = 0.12$; range: $[0.03, 0.27]$; $p_{fisher} < 10^{-4}$). Interestingly, the relationship between temporal SFC variance and Hurst exponents was non-linear and heteroscedastic (Breusch-Pagan test: $p_{fisher} < 10^{-4}$). Accordingly, we used a quadratic regression and found that the highest temporal variance in SFC occurred for middle Hurst exponent values (Fig. 6D; Supplemental Fig. 15D; mean $\beta_{stand}$ for quadratic term = $-0.47$; range: $[-1.13, 0.26]$; $p_{fisher} < 10^{-4}$); this finding points towards temporal fluctuations in SFC reaching a plateau with increasing levels of relative synaptic inhibition. Lastly—and in contrast to the atlas-based analyses—there was a significantly positive association between SFC and the Hurst exponent across the different temporal windows (average Spearman's $\rho$ across subjects and across brain regions: 0.03; $\rho$ range across subjects: $[0.01, 0.08]$; $p_{fisher\ (FDR\text{-}corrected)} < 10^{-4}$), indicating that SFC and EI-ratio co-fluctuate over short periods of time (i.e., the duration of

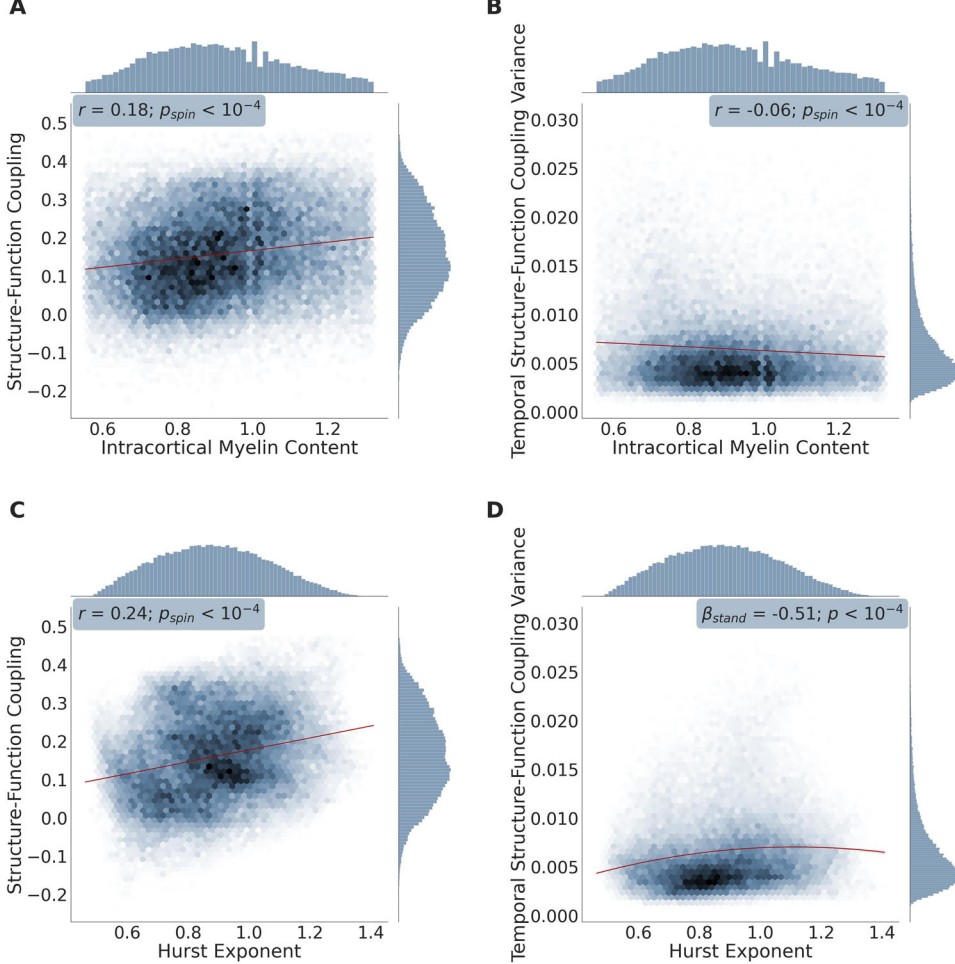

**Fig. 6 | High density plots between the variables of interest: voxel-based analysis – representative subject shown.** High density plots showing the association between each cortical voxel's: mean structure-function coupling and intracortical myelin content estimated by the T1-weighted/T2-weighted signal intensity ratio (**A**), mean temporal structure-function coupling variance and intracortical myelin content (**B**), mean structure-function coupling and the Hurst exponent of the functional signal time series (**C**), and mean temporal structure-function coupling variance and the Hurst exponent of the functional signal time series (**D**). For plots (**A**), (**B**), and (**C**), a linear regression was fit along with a 95% confidence interval (shown in red); correlation coefficients (two-tailed Spearman's $\rho$: $r$), $p$-values

corresponding to the spatial permutation test ($p_{spin}$), and histograms corresponding to each variable are displayed. In plot (**D**), a quadratic regression was fit along with a 95% confidence interval (shown in red); the standardized $\beta$ coefficient and bootstrapped $p$-value corresponding to the quadratic regression mentioned in the voxel-based analysis component of our Results section: 'Biological Correlates of Structure-Function Coupling: Whole-brain perspective,' are also reported. Data shown in this figure were obtained from a representative subject that was analyzed using our voxel-based connectivity approach. Scatterplot versions of the above plots are shown in Supplemental Fig. 15. $n = 71,561$ voxels in all panels. Source data are provided as a Source Data file.

the fMRI scan), when examined under the finer spatial scale defined by our voxel-based analysis.

We next re-examined the above relationships using multiple linear regression models. Cortical voxels' SFC was independently and positively correlated with intracortical myelin content (mean $\beta_{stand} = 0.07$; range: [0.02, 0.11]; $p_{fisher} < 10^{-4}$; mean $VIF = 1.06$; range: [1.04, 1.08]) and with the Hurst exponent (mean $\beta_{stand} = 0.12$; range: [0.05, 0.25]; $p_{fisher} < 10^{-4}$; mean $VIF = 1.02$; range: [1, 1.05]), even after adjusting for the effects of the other biological marker and the voxels' placement along the cortical hierarchy. After additionally including a non-linear (Hurst exponent squared) component in the multiple regression model to account for the non-linear relationship between temporal SFC variance and the Hurst exponent, we found that temporal SFC variance was independently and negatively correlated with intracortical myelin content (mean $\beta_{stand} = -0.05$; range: [−0.09, −0.01]; $p_{fisher} < 10^{-4}$, mean $VIF = 1.06$; range: [1.04, 1.08]) and with the squared Hurst exponent (mean $\beta_{stand} = -0.46$; range: [−0.82, 0.18]; $p_{fisher} < 10^{-4}$), after adjusting for the other biological marker of interest and the principal gradient assignment.

## Biological correlates of structure-function coupling: regional perspective

To further decipher *how* SFC is dynamically regulated within different networks along the cortical hierarchy, we next combined elements from the previous two sections to investigate the dynamic relationship between SFC, temporal SFC variance, intracortical myelination, and Hurst exponents across different cyto-architectonic systems of varying laminar differentiation. Specifically, instead of applying multiple regression models at the whole-brain level as we did in the previous section, here we separately applied them to each von Economo/Koskinas-inspired cyto-architectonic class.

We begin with the cyto-architectonic class that displayed the highest SFC: the granular type. We observed a significant positive association between SFC (dependent variable) and the Hurst exponent but not with intracortical myelin content, after adjusting for the effects of the other biological marker (Table 1A). In the parietal and frontal types, we observed a significant positive association between SFC and the Hurst exponent as well as the intracortical myelin content (Table 1A). Within the agranular cyto-architectonic class, we observed

**Table 1 | Atlas-based multiple linear regression analyses**

| | Intracortical Myelin | | | Hurst Exponent | | | |
|---|---|---|---|---|---|---|---|
| **A. Structure-Function Coupling** | | | | | | | |
| Cortical Type | $\beta_{stand}$ | 95% BCI | Bootstrapped p-value (FDR) | $\beta_{stand}$ | 95% BCI | Bootstrapped p-value (FDR) | VIF |
| Granular | 0.144 | [0.099, 0.108] | 0.370 | 0.758 | [0.765, 0.771] | $6.7 \times 10^{-4}$ | 1.00 |
| Polar | 0.332 | [0.256, 0.267] | 0.370 | −0.124 | [−0.112, −0.102] | 0.651 | 1.26 |
| Parietal | 0.407 | [0.402, 0.405] | $<10^{-4}$ | 0.567 | [0.562, 0.565] | $<10^{-4}$ | 1.00 |
| Frontal | 0.423 | [0.423, 0.426] | $<10^{-4}$ | 0.326 | [0.322, 0.324] | $<10^{-4}$ | 1.06 |
| Agranular | 0.465 | [0.465, 0.469] | $<10^{-4}$ | 0.121 | [0.121, 0.126] | 0.449 | 1.11 |
| **B. Temporal Structure-Function Coupling Variance** | | | | | | | |
| Granular | 0.446 | [0.411, 0.417] | 0.051 | −0.604 | [−0.596, −0.590] | 0.003 | 1.00 |
| Polar | −0.485 | [−0.496, −0.491] | 0.004 | −0.446 | [−0.446, −0.441] | 0.002 | 1.26 |
| Parietal | −0.184 | [−0.187, −0.184] | 0.051 | −0.370 | [−0.373, −0.368] | 0.004 | 1.00 |
| Frontal | −0.135 | [−0.136, −0.134] | 0.051 | −0.261 | [−0.258, −0.255] | $5 \times 10^{-4}$ | 1.06 |
| Agranular | −0.262 | [−0.259, −0.256] | 0.004 | −0.570 | [−0.559, −0.554] | $<10^{-4}$ | 1.11 |

Results corresponding to the atlas-based analyses discussed in section: 'Biological Correlates of Structure-Function Coupling: Regional perspective.' $\beta_{stand}$: standardized $\beta$ coefficient; 95% BCI: 95% bootstrapped standardized $\beta$ coefficient confidence interval; Bootstrapped p-value (FDR): bootstrapped p-value adjusted for multiple comparisons (two-tailed test; false discovery rate [FDR]: Benjamini-Hochberg method); VIF: Variance Inflation Factor.

that SFC was positively correlated only with intracortical myelin content but not with the Hurst exponent, within the same regression model (Table 1A). Taking these results together, we notice a distinct pattern as we transition from granular to agranular cortical regions: a gradual shift from the Hurst exponent to intracortical myelin content as being the principal predictor of SFC (as supported by the numerical changes in the standardized $\beta$ and false discovery rate-adjusted p-values: Table 1A; Fig. 7). Importantly, this pattern was also reproduced across our Supplemental Analyses (Supplemental Material: Supplemental Analyses 1, 2, 3; Supplemental Tables 9, 14, 19, 24). Notably, the cortical type with the lowest SFC and relatively high levels of granularization—the polar type—was an exception to this rule, with SFC not being significantly correlated with either intracortical myelin content or the Hurst exponent (Table 1A; Supplemental Material: Methodological Considerations and Study Limitations).

Interestingly, the temporal SFC variance correlated significantly with both intracortical myelin content and the Hurst exponent, across most cyto-architectonic classes. It was, however, more persistently dependent upon the Hurst exponent across all cortical types, after adjusting for the effects of intracortical myelin content (Table 1B; Supplemental Tables 9, 14, 19, 24).

Using the voxel-based approach produced similar results. Specifically, within the granular type, we again observed a positive independent correlation between SFC and the Hurst exponent but not with intracortical myelin content (Table 2A). Within the polar and parietal types, intracortical myelination's effect size in predicting SFC increased; SFC independently correlated with both myelin content and the Hurst exponent (Table 2A). Further, SFC independently correlated with both biological markers within the frontal and agranular types, with intracortical myelination's predictive effect of SFC surpassing that of the Hurst exponent within the frontal type (Table 2A). Thus, these voxel-level results support, once again, the notion of a gradual transition from granular to less granular cortical regions in the degree to which the Hurst exponent (and therefore EI-ratio) and intracortical myelination predict SFC.

Similar to the atlas-based results, temporal SFC variance displayed a stronger dependence upon the Hurst exponent as its predictor across all cyto-architectonic classes. Specifically, temporal SFC variance was independently correlated with the squared Hurst exponent, after adjusting for the effects of the Hurst exponent and intracortical myelin content in each cyto-architectonic class (Table 2B). In the voxel-based analyses, intracortical myelin content was also independently correlated with temporal SFC variance across all classes with a lower,

however, overall effect size compared to that of the Hurst exponent (Table 2B).

## Discussion

In order to better understand how structure shapes and constrains function in the human brain, recent work has introduced the notion of SFC, a metric quantifying how strongly a brain region's functional connectivity with other brain regions mirrors its structural connectivity. SFC has often been found to capture more than just the sum of its parts: regional variations in SFC can more accurately predict differences in cognitive performance as well as track neurological disease symptomatology and duration, than structural or functional connectivity alone[6,16,17,19,47,48]. Hence, we sought to understand *how* SFC varies across different brain regions within the healthy human brain, as well as *why*—what underlying biological factors mediate such variation?

We specifically addressed three complementary aims. First, we assessed changes in SFC and temporal SFC variance across the sensory-association gradient. Second, we examined whether the spatial expressions of SFC and its temporal variance were correlated with those of intracortical myelination and EI-ratio across the cortex. Third, we analyzed the association of SFC and its temporal variance with both intracortical myelination and EI-ratio, within different cyto-architectonic cortical types, in order to investigate how SFC is dynamically regulated at the level of individual networks. To ensure the generalizability of our results, we analyzed neuroimaging data obtained from two independent groups of healthy participants using six complementary processing pipelines: atlas-based approaches capitalizing on two different brain parcellation schemes, and a voxel-based approach of uncommonly high resolution wherein each subject's cortical voxel was designated as a stand-alone brain region.

Addressing our first aim, we asked to what extent SFC captures macroscale functional and microscale cyto-architectonic organization principles, and we answered that question by examining regional variations in SFC and its temporal variance across the cortical hierarchy. Across all processing pipelines, we found an overall increase in temporal SFC variance along the unimodal (sensory)-transmodal (association) hierarchy, where the highest deviations from the mean occurred in the limbic regions. This finding largely parallels results from a recent study using a different definition of temporal SFC variance (see Supplemental Material: Methodological Considerations and Study Limitations), also demonstrating that a region's ability to dynamically fluctuate its SFC over time depends on its location along the

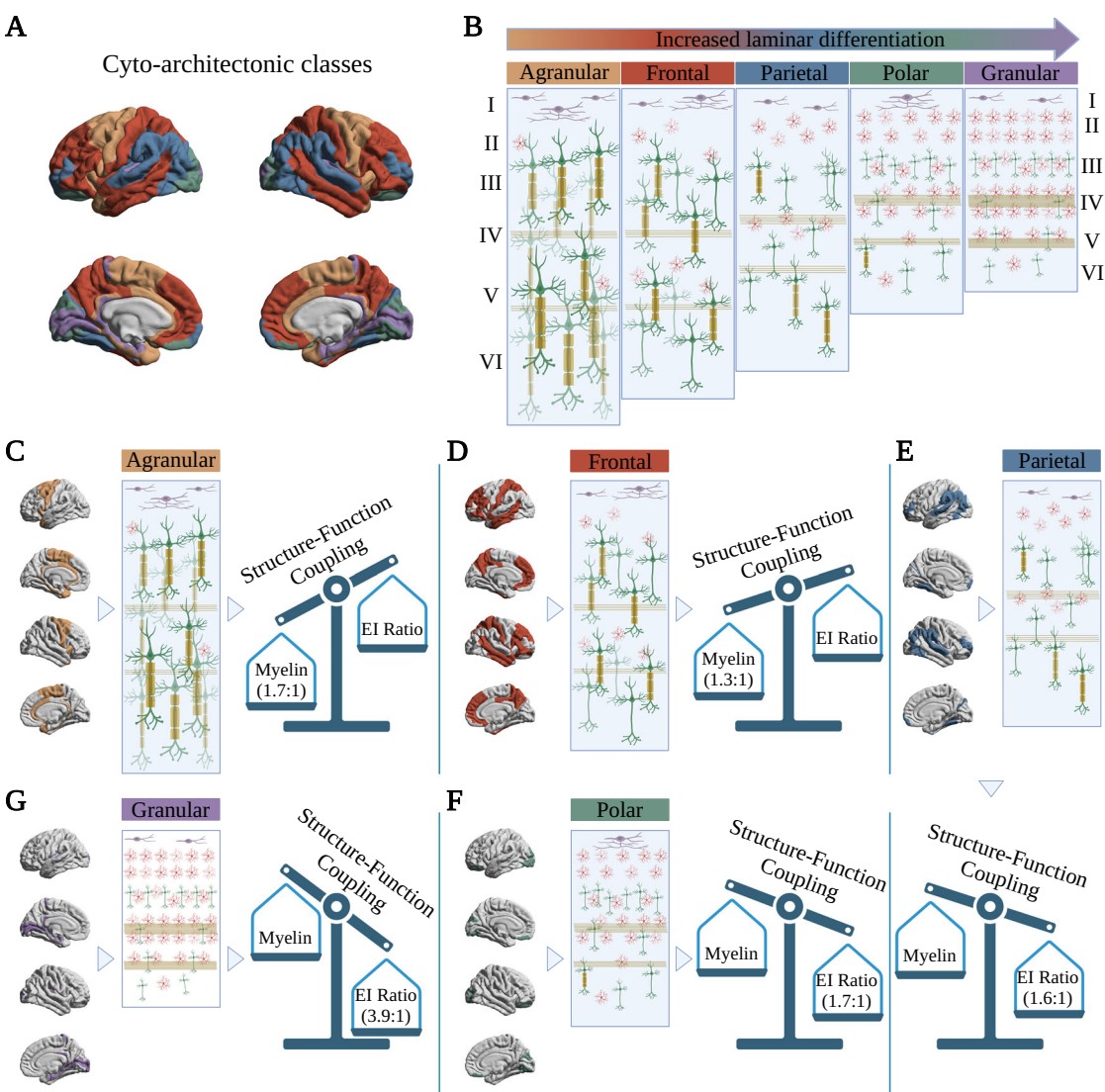

**Fig. 7 | Shift from intracortical myelination to excitation-inhibition ratio as the principal predictor of structure-function coupling, when transitioning from agranular to granular cortical regions: an illustration. A** Parcellation of the cortex into the 5 von Economo/Koskinas-inspired cyto-architectonic classes: agranular (orange), frontal (red), parietal (blue), polar (green), and granular (purple). **B** Schematic illustration of the types and distribution of cells expected to occupy the numbered cortical layers within each cyto-architectonic class. Purple cells represent Cajal-Retzius neurons; red star-shaped cells represent stellate cells; green cells with triangular somata represent pyramidal cells; myelin sheaths are shown in dark yellow; and the stripes across each column represent the outer (layer IV) and inner (layer V) stripes (or bands) of Baillarger (myelinated fibers arising mostly from the thalamus) in dark yellow. While transitioning from granular to agranular cortical types, we notice an increase in axonal myelination, an increase in the number and size of pyramidal neurons, and a decrease in the number of small stellate neurons. **C–G** The left side of each panel displays the brain regions corresponding to each cyto-architectonic class. The middle figure corresponds to the expected cellular distribution and composition of each class as defined in panel (B). The third figure on the right represents a visual scale wherein the contributions of intracortical myelination and excitation-inhibition (EI) ratio (in the form of the Hurst exponent) in predicting structure-function coupling are "weighed" against each other. For each cyto-architectonic class, such "weight" was determined by computing the ratio between the $\beta_{stand}$ coefficient corresponding to the Hurst exponent and the $\beta_{stand}$ coefficient corresponding to intracortical myelin across our three main processing pipelines: the Schaefer 400 atlas-based (Table 1), the HCP multi-modal atlas-based (Supplemental Table 9), and the voxel-based (Table 2) analyses. This generated three ratio values for each cortical type which were then averaged, to generate one representative ratio value per cortical type indicating how much−on average−one variable contributes more than the other in predicting structure-function coupling. The averaged ratio is shown underneath the 'dominant' variable's name across each scale. Created with BioRender.com.

unimodal-transmodal hierarchy[49]. Put together, these observations could indeed reflect the inherently increased functional connectivity variability found in heteromodal association cortices, compared to unimodal cortices[50].

Moreover, SFC consistently and gradually decreased along the unimodal-transmodal hierarchy, in agreement with previous work on the field[1,4,6,21–23,49]. Specifically, in our analyses, SFC decreased while transitioning from granular cortical areas with pronounced laminar organization (i.e., granularization), such as the primary sensory regions, to areas with progressively diminishing laminar differentiation, namely the parietal, frontal, and finally agranular cyto-architectonic cortical types, reaching its lowest value in the agranular limbic regions. The sole deviation from this pattern was found in the polar cortical type, which had a significantly lower SFC and higher temporal SFC variance compared to the remaining 4 cyto-architectonic classes, despite its relatively high granularization. This result can be usefully interpreted from a functional perspective: the polar cortical type predominantly comprises higher-order visual association areas and a large portion of the transmodal orbitofrontal cortex[44]. The latter region flexibly encodes reward and punishment

**Table 2 | Voxel-based multiple linear regression analyses**

**A. Structure-Function Coupling**

| Cortical Type | Intracortical Myelin | | | Hurst Exponent | | | Mean VIF | Range VIF |
|---|---|---|---|---|---|---|---|---|
| | Mean $\beta_{stand}$ | Range $\beta_{stand}$ | Fisher's p-value (FDR) | Mean $\beta_{stand}$ | Range $\beta_{stand}$ | Fisher's p-value (FDR) | | |
| Granular | 0.01 | [−0.02, 0.04] | 0.084 | 0.05 | [−0.09, 0.35] | $<10^{-4}$ | 1 | [1.00, 1.01] |
| Polar | 0.03 | [−0.05, 0.12] | $<10^{-4}$ | 0.18 | [−0.16, 0.57] | $<10^{-4}$ | 1.05 | [1.01, 1.07] |
| Parietal | 0.02 | [−0.07, 0.09] | $<10^{-4}$ | 0.05 | [−0.07, 0.17] | $<10^{-4}$ | 1 | [1.00, 1.01] |
| Frontal | 0.13 | [0.05, 0.18] | $<10^{-4}$ | 0.11 | [0.01, 0.29] | $<10^{-4}$ | 1.02 | [1.01, 1.05] |
| Agranular | 0.16 | [0.06, 0.24] | $<10^{-4}$ | 0.23 | [0.05, 0.42] | $<10^{-4}$ | 1.02 | [1.00, 1.08] |

**B. Temporal Structure-Function Coupling Variance**

| Cortical Type | Intracortical Myelin | | | Hurst Exponent² | | | Mean VIF | Range VIF |
|---|---|---|---|---|---|---|---|---|
| | Mean $\beta_{stand}$ | Range $\beta_{stand}$ | Fisher's p-value (FDR) | Mean $\beta_{stand}$ | Range $\beta_{stand}$ | Fisher's p-value (FDR) | | |
| Granular | −0.002 | [−0.03, 0.06] | 0.005 | −0.10 | [−2.37, 2.00] | $<10^{-4}$ | 1.01 | [1.00, 1.01] |
| Polar | −0.05 | [−0.13, 0.09] | $<10^{-4}$ | 0.27 | [−2.41, 2.38] | $<10^{-4}$ | 1.06 | [1.01, 1.08] |
| Parietal | −0.09 | [−0.15, −0.03] | $<10^{-4}$ | −0.68 | [−1.26, −0.15] | $<10^{-4}$ | 1.01 | [1.00, 1.01] |
| Frontal | −0.12 | [−0.17, −0.05] | $<10^{-4}$ | −0.48 | [−1.23, 0.58] | $<10^{-4}$ | 1.02 | [1.01, 1.05] |
| Agranular | −0.03 | [−0.13, 0.05] | $<10^{-4}$ | 0.09 | [−0.80, 1.43] | $<10^{-4}$ | 1.02 | [1.01, 1.08] |

Results corresponding to the voxel-based analyses discussed in section: 'Biological Correlates of Structure-Function Coupling: Regional perspective.' Mean $\beta_{stand}$: the mean standardized $\beta$ coefficient across the 9 subjects; Range $\beta_{stand}$: [min, max] range of the standardized $\beta$ coefficient across the 9 subjects; Fisher's p-value (FDR): Fisher's p-value adjusted for false discovery rate [FDR] using the Benjamini-Hochberg method (two-tailed test); Mean VIF: the mean Variance Inflation Factor across the 9 subjects; Range VIF: [min, max] range of VIF values across the 9 subjects; Hurst Exponent²: Hurst exponent squared.

values of stimuli[51], supporting the notion that higher-order association areas heavily involved in emotional regulation have particularly low 'static' SFC that fluctuates markedly across time.

Furthermore, studies using definitions of SFC other than the correlational approach utilized in this work have also found a heterogeneous decoupling between structure and function across the sensory-association hierarchy. Such definitions have quantified SFC by invoking (i) spectral graph theory, where a brain region's functional brain activity (typically the blood oxygen level-dependent [BOLD] signal) is expressed as a weighted linear combination of the harmonic components of the brain's structural connectome (i.e., as defined by its eigendecomposition); structure-function decoupling can then be assessed as the ratio between the higher spatial frequency 'decoupled' and lower spatial frequency 'coupled' portions of the spectrum[22,52], and (ii) linear regression modeling approaches, where a brain region's SFC is assessed by how well its empirically-defined functional connectivity can be predicted by linear models incorporating markers of structural organization, such as Euclidean distance, shortest path length, and communicability, obtained from the structural connectome[21,49]. Such complementary approaches can be particularly informative in deciphering the spatial and topological attributes of the structural connectome most relevant in mediating SFC.

Addressing our second aim, we asked *why* SFC regionally varies across the brain, and we answered that question by examining whether the heterogeneous spatial expressions of SFC and temporal SFC variance across the cortex were correlated with that of intracortical myelination and the functional time series' Hurst exponent, which represents a proxy for EI-ratio. Across both atlas- and voxel-based analyses, we found that the functional connectivity patterns of heavily myelinated brain regions strongly reflect their underlying structural connectivity patterns; increased myelination also constrained how much the correlation between structure and function deviated from its mean value over time. Similarly, the functional connectivity of brain regions characterized by increased levels of relative inhibition, in the form of an increased Hurst exponent, largely mirrored the strength of the underlying anatomical connectivity; regions with increased levels of relative inhibition exhibited lower SFC variance over time, as well. Notably, in our atlas-based analyses, the EI-ratio accounted for 24-50% of the correlation between intracortical myelin and temporal SFC variance.

Our results highlight the critical role that both myelination and EI balance play in regulating how much and how often the BOLD signal propagation patterns deviate from the underlying anatomical backbone. Increased levels of myelination have been reported to suppress the formation of new axonal tracts and synapses[27,53], thus potentially constraining the emergence of functional signals that deviate from structural paths. In turn, the heavy myelination observed in some brain regions, such as the primary sensory and motor cortices, could support these regions' functional specialization. Lower levels of myelination, on the other hand, allow for greater functional signal variability and continuous neuronal remodeling to take place at various time scales[29,31,54], enabling the emergence of functional dynamics that can more richly diverge from structural connectivity. The enhanced affinity for neuroplasticity within lightly myelinated transmodal regions could thus foster the emergence of flexible functional dynamics characteristic of adaptive behavior and learning.

In parallel, brain regions predominantly characterized by inhibitory regimes would also be expected to display functional dynamics that deviate less from the underlying structural paths. Indeed, neuronal assemblies characterized by increased relative inhibition—whether due to decreased synaptic excitation or increased inhibition—favor BOLD activity of decreased signal amplitude[55,56], a decreased plateau phase following the initial peak (i.e., faster response adaptation)[57], and lower overall baseline neuronal firing rates[57-59]. Additionally, inhibition acts as a stabilizing agent of cortical activity, constraining any aberrant

amplification of neuronal firing arising from recurrent excitation[60,61]; synaptic inhibition can also spatially and temporally constrain the spread as well as sharpen the evoked BOLD signals in response to sensory input[62–66].

Addressing our third and last aim, we asked whether the relation between SFC and the two biological substrates of interest changes across the sensory-association hierarchy, and specifically within cortical regions of varying cyto-architectonic properties. Pioneering work in the early 20th century led to the parcellation of the cerebral cortex into 5 distinct structural types based on cellular morphology, cyto-architectonic properties, and cortical thickness: granular, polar, parietal, frontal, and agranular[39,43,44,67,68]. At one end of the spectrum, the thin granular cortex (also known as koniocortex) is distinguished by well-defined, highly-developed cortical layers II and IV, and houses densely packed small stellate and pyramidal cells, collectively referred to as granule neurons[39,43,67]; these cells typically have short axons projecting locally within the cortex and very small multi-polar cell bodies, with a cell body diameter ≲10 μm[69]. Functionally, the granular cortex encompasses primary sensory (visual, auditory, and somatosensory) areas and parts of the parahippocampal gyrus[43,44,68]. At the opposite end of the laminar differentiation spectrum, the thicker agranular cortex has particularly thin or absent granular laminae II and IV, and predominantly houses large pyramidal neurons spanning multiple cortical layers[43,69]. Although typically associated with motor cortices, the agranular cortex also encompasses limbic regions such as the anterior fronto-insular and cingulate cortices[68,70]. The remaining cortical types (polar, parietal, frontal) capture intermediate, progressively decreasing levels of granularization with generally increasing neuronal cell sizes[43].

Across processing pipelines, we observed a gradual shift when traversing from granular to agranular cortical types, from the EI-ratio (i.e., Hurst exponent) to intracortical myelin content as being the principal predictor of SFC. Given the aforementioned differences in the cyto-architectonic properties of the cortical types, this finding is intuitive: granule cells predominantly found in the granular cortical regions are typically unmyelinated, mainly due to their small axonal diameters (≲0.3 μm[68,71,72]; neurons in the central nervous system with axonal diameters ≲0.3 μm are *usually* unmyelinated[73–76]) and the increased metabolic cost that would be required to myelinate such short axons projecting locally, without necessarily an accompanying enhancement of signal conduction velocity[76,77]. In turn, the lack of myelin sheath directly exposes these axons to the extracellular space, making them particularly susceptible to subthreshold excitability changes[77]. Therefore, the correspondence between structural and functional connectivity within cortical regions characterized by increased levels of granule cells would be expected to be more dependent upon fluctuations in excitation and inhibition, rather than intracortical myelin levels.

On the other hand, the large pyramidal neurons prominently occupying cortical areas with decreased levels of granularization are highly myelinated[78,79]; small granule cells are significantly sparser in these layers. Following the same line of reasoning as above, it would thus be expected that the contribution of intracortical myelination levels in these regions in predicting their macroscale SFC would significantly increase. Interestingly, in our voxel-based analyses both intracortical myelination and EI-ratio played a significant role in predicting SFC in the agranular cortical regions. In the coarser atlas-based analyses, however, this effect was averaged out, leaving only intracortical myelination as the primary predictor of SFC in that cortical type. This finding could indicate that on the macroscale level captured by the atlas-based parcellations, intracortical myelination cumulatively plays a more significant role than EI balance in shaping the coupling between structure and function in agranular cortical regions.

Finally, intracortical myelination and EI-ratio together played a significant role in shaping temporal fluctuations in SFC. Overall, the EI-ratio had a larger effect size in predicting moment-to-moment SFC variance than intracortical myelination, and consistently correlated with the amount of moment-to-moment SFC variance across each cyto-architectonic class in both atlas- and voxel-based analyses. This finding is not surprising given how the balance between excitation and inhibition also fluctuates on a moment-to-moment basis[80]. Intracortical myelination, on the other hand, does not typically fluctuate on such short timescales in the resting brain, and it is thus likely to constrain how often the BOLD signal propagation patterns can deviate from the anatomical backbone on a slower time scale.

Collectively, the extent to which the spontaneous activity of a brain region is tethered to the underlying white matter projections is evidently shaped by the regional intracortical myelin content and EI-ratio. Intracortical myelination and neuromodulation, however, are highly multi-faceted properties, each representing the concerted sum of other biological properties. Myelination patterns, for instance, rely upon glial-neuronal interactions[81,82] as well as genetic and environmental influences[83,84], and have been shown to extend well into the third decade of life[84]. Moreover, neuronal excitation and inhibition patterns are mediated by the release of excitatory (e.g., glutamate) or inhibitory (i.e., gamma-aminobutyric acid) neurotransmitters, and modulated by the activity of major regulatory systems in the central nervous system, such as the dopaminergic, noradrenergic, serotonergic, and cholinergic systems, at any given time. Therefore, it would be critical to examine in future studies how each of these individual facets of neurobiology sculpts the dynamic relationship between structural and functional connectivity in the human brain, at rest or during a task, in health or disease.

In this study, we examined the regional dependence between structure and function across complementary cortical hierarchies, and aimed to identify the biological factors that mediate such coupling in the human brain. We assessed the correlation between structure and function using atlas- as well as voxel-based connectivity, capturing the underlying anatomy and dynamics in marked detail. Our findings were consistent across all processing pipelines and cohorts employed, and are three-fold: (1) SFC and its temporal variance respectively decrease and increase across the unimodal-transmodal and granular-agranular gradients, (2) increased intracortical myelination and lower EI-ratio are associated with a more rigid coupling between structure and function and restricted moment-to-moment SFC fluctuations, and (3) there is a gradual shift from EI-ratio to intracortical myelination as being the principal predictor of SFC when traversing from granular to agranular cortical types; EI-ratio appears to be the principal predictor of temporal SFC variance within each cyto-architectonic type. Overall, our results identify regional intracortical myelination and EI balance as factors that synergistically shape how strongly coupled the functional expression of the human cortex is to its underlying anatomical connectivity. Such an explanatory relationship could provide invaluable insight into the aberrant coupling between structure and function in neurological and psychiatric disorders characterized by demyelination and/or EI imbalances.

## Methods

### Datasets

**Human Connectome Project.** A sample of 100 unrelated healthy subjects (54% female; mean age = 29.1 ± 3.7 years; age range = 22–36 years) was drawn from the HCP dataset, as publicly provided by the HCP1200 subjects data release[85]. Subjects within this sample were scanned on a customized Siemens "Connectome" Skyra 3 T scanner (32-channel Siemens head coil) and underwent high-resolution 3 T MRI, including T1-weighted (3D Multi-echo Magnetization–Prepared Rapid Gradient Echo [MEMPRAGE] sequence; voxel size: 0.7 mm isotropic; repetition time [TR]: 2400 ms; echo time [TE]: 2.14 ms), T2-weighted (3D sampling perfection with application-optimized contrasts by using flip angle evolution [SPACE] sequence; voxel size:

0.7 mm isotropic; TR: 3200 ms; TE: 565 ms), resting-state fMRI (gradient-echo echo-planar imaging [EPI] sequence; four runs; 1200 volumes/run, 14:33 min:sec each; voxel size: 2 mm isotropic; TR: 720 ms; TE: 33.1 ms), and high angular resolution diffusion imaging (spin-echo planar imaging sequence; voxel size: 1.25 mm isotropic; TR: 5520 ms; TE: 89.5 ms; max $b$-value: 3000 s/mm$^2$; 270 non-colinear directions; 18 $b0$ acquisitions) sequences[86,87]. Informed consent was obtained from all subjects, and the procedures were approved by the Washington University Institutional Review Board.

**Penn sample.** Healthy individuals ($n = 14$; 78.6% female; mean age = 22.8 ± 3.2 years; age range = 18–28 years) were prospectively enrolled at the University of Pennsylvania between November 16, 2016 and May 19, 2018, and recruited from the local community. Subjects within this sample were scanned using a Siemens Magnetom Prisma 3 T scanner (64-channel head/neck coil) and underwent high-resolution 3 T MRI, including T1-weighted (3D MEMPRAGE sequence; voxel size: 0.9 mm isotropic; TR: 2500 ms; TE: 2.18 ms), T2-weighted (3D SPACE sequence; voxel size: 0.9 mm isotropic; TR: 3200 ms; TE: 565 ms), resting-state functional MRI (two runs; 1200 volumes/run, 20:07 min:sec each; voxel size: 3 mm isotropic; TR: 500 ms; TE: 25 ms), and diffusion spectrum imaging (DSI; voxel size: 1.8 mm isotropic; TR: 4300 ms; TE: 102 ms; max $b$-value: 5000 s/mm$^2$; 731 directions; 22 $b0$ acquisitions) sequences. Informed consent was obtained from all subjects, and the procedures were approved by the University of Pennsylvania Institutional Review Board.

### Processing pipelines

In order to analyze our two samples and test the reproducibility of our results, we utilized six complementary processing pipelines (Fig. 1): (1) atlas-based approaches wherein each subject's cortex was parcellated using two common brain atlases; structural and functional connectivity values were estimated between all region pairs defined by these coarse-grained atlases, and (2) a significantly more fine-grained voxel-based approach, wherein each subject's cortical voxel was designated as a stand-alone brain region; for each participant, structural and functional connectivity values were estimated between all possible pairs of their cortical voxels. For this purpose, custom scripts were written using bash shell scripting (version 3.2.57).

**Atlas-based approach.** This approach was used to analyze both the HCP and Penn samples. The results corresponding to the HCP sample can be found in the Results section of the Manuscript and in the Supplemental Material, under Supplemental Analyses 1 and 2. The results corresponding to the Penn sample can be found in the Supplemental Material, under Supplemental Analysis 3 (Fig. 1).

For each subject, we first estimated their structural connectivity, then their functional connectivity, and lastly their SFC. The different types of connectivity were quantified using two commonly used brain atlases: the functionally-inspired Schaefer atlas[42] (400 cortical parcels) and the HCP multi-modal atlas[45] (360 cortical parcels).

**Structural connectivity.** The subjects' diffusion scans were first minimally pre-processed using the HCP consortium pipelines (https://github.com/Washington-University/HCPpipelines), which included applying $b0$ intensity normalization, correcting for EPI distortion, Eddy currents, subject motion, and gradient nonlinearity, and registering them to the subject's native T1-weighted anatomical scan[88]. Further processing of diffusion data was carried out using the *MRtrix3* toolbox[89]. Multi-shell, multi-tissue constrained spherical deconvolution was first performed to generate fiber orientation densities. Anatomically constrained probabilistic tractography was then applied using a second-order integration over fiber orientation distributions method, to more accurately track fibers through crossing regions[90]. An initial whole-brain tractogram containing ten

million streamlines was generated for each subject, which was then corrected by assigning each streamlines a weight to reduce known biases in tractography data and better match the diffusion properties of the empirical data (SIFT2 approach)[6,91]. Each subject's refined whole-brain tractogram was finally mapped to each parcellated brain atlas (Schaefer and HCP multi-modal) that had been registered to the subject's native space, to produce two subject-specific, symmetric, weighted structural connectomes (Schaefer atlas: 400 ROIs × 400 ROIs, HCP multi-modal atlas: 360 ROIs × 360 ROIs). In each connectome, the structural connectivity between any two given brain regions (i.e., network edge) was defined as the SIFT2-weighted sum of the streamlines connecting these two regions divided by the sum of the regions' gray matter volumes[6].

**Functional connectivity.** Similar to the diffusion scans, the resting-state fMRI scans were also minimally pre-processed using the HCP consortium pipelines. These pre-processing pipelines included correcting for gradient distortion, subject motion, and EPI image distortion, as well as intensity normalization and registration of the functional scans to the standard MNI space[88]. The resulting functional signal time series were accurately aligned across subjects using an areal feature-based cross-subject alignment method (MSMAll)[45] and further denoised from artifact and linear trends using an independent component analysis and hierarchical fusion of classifiers approach (sICA + FIX)[92].

In the HCP sample, pre-processed time series corresponding to each run were then demeaned and normalized. Then two complementary approaches were followed: First, all four runs (1200 volumes per run) were averaged into one run (1200 volumes) for each subject, using the *Connectome Workbench* toolbox; the time series corresponding to the voxels within broader brain regions were averaged to produce matrices of size: number of ROIs × 1200 volumes (Schaefer atlas: 400 ROIs × 1200 volumes, HCP multi-modal atlas: 360 ROIs × 1200 volumes), for each subject. The results corresponding to this approach can be found in the Results section of the manuscript and in Supplemental Analysis 1, found in the Supplemental Material. In the second approach, instead of averaging the demeaned and normalized pre-processed time series corresponding to the four runs into one average run (1200 volumes), we concatenated all four runs across time (1200 volumes × 4 runs) for each subject. The time series corresponding to the voxels within broader brain regions were then averaged to produce matrices of size: number of ROIs × 4800 volumes (Schaefer atlas: 400 ROIs × 4800 volumes, HCP multi-modal atlas: 360 ROIs × 4800 volumes), for each subject. The results corresponding to this approach can be found in the Supplemental Material, under Supplemental Analyses 2A and 2B.

'Static' functional connectivity matrices for each atlas were computed by calculating the Pearson's correlation between the average signal time series of any two given brain regions; each entry in the functional connectome is equal to the Pearson's correlation coefficient between the activity time series of the regions corresponding to the matrix element's row and column. In order to examine how each subject's functional connectivity changes across time, we split each atlas' signal time series matrix into 20 continuous non-overlapping time windows. This procedure allowed us to generate 20 'temporally-contiguous' functional connectomes per subject.

**Structure-function coupling.** For each atlas, the SFC of each subject's brain region was defined as the Pearson's correlation coefficient between the row corresponding to that region in the structural connectome and the row corresponding to that region in the 'static' functional connectome (Schaefer atlas: 1 × 400 ROIs, HCP multi-modal atlas: 1 × 360 ROIs), after excluding the self-connection and any other entries where either the regional structural or functional connectivity was equal to zero.

To examine how much SFC deviates from its mean value over time, we also computed its moment-to-moment variance throughout the duration of the resting-state fMRI scan. Here, instead of computing one 'static' SFC value for each brain region—as was done in the analyses described earlier—we computed an SFC value for each one of the 'temporally-contiguous' functional connectivity matrices defined in the 'Functional Connectivity' section above, and the underlying structural connectivity matrix (once again, self-connections and entries where either the regional structural or functional connectivity was equal to zero were excluded). Since we used 20 non-overlapping time windows, we ended up with 20 SFC values for each brain region, for each subject. Each brain region's temporal SFC variance was then defined as the variance across their corresponding 20 'temporally-contiguous' SFC values (Supplemental Material: Eq. 1).

The two analyses just described produced two metrics that quantified the nature of coupling between a brain region's structural and functional connectivity: (i) a 'static' SFC: a metric indicating how strongly coupled functional connectivity is to the underlying structural connectivity, overall, and (ii) the temporal SFC variance: a metric indicating how much SFC deviates from its mean value over time.

**Intracortical myelination.** Intracortical myelination was assessed using a previously validated T1-weighted/T2-weighted ratio approach[88], the scripts for which were provided by the latest HCP1200 subjects data release. There are three main pipelines used by the HCP consortium to compute intracortical myelin surface maps for each subject, and we describe them briefly here: within the first *Pre-FreeSurfer* pipeline, the T1- and T2-weighted sequences are first corrected for gradient and readout distortions. The undistorted T2-weighted image is then registered to the undistorted T1-weighted image, after which they are both bias-field corrected. In the subsequent *FreeSurfer* pipeline, the undistorted bias-corrected T1-weighted image in each subject's native volume space is input into the *FreeSurfer* software suite (https://www.surfer.nmr.mgh.harvard.edu)[93] to generate highly accurate white matter and pial cortical surfaces. The T1-weighted image is then intensity normalized, and contrast signal intensity information from the undistorted bias-corrected T2-weighted image is used to update the pial surfaces such that they exclude dura and blood vessels. Registration of the T2- to the T1-weighted image is fine-tuned even further throughout this pipeline, using *FreeSurfer*'s boundary-based registration tool by incorporating information from the reconstructed surfaces. During the last *PostFreeSurfer* processing pipeline, the *FreeSurfer*-derived white and pial surfaces, along with other morphometric measurements such as cortical thickness, curvature, and folding patterns, are used to define a highly accurate, high-resolution cortical ribbon volume. The T1-weighted image is then divided by the aligned T2-weighted image—a mathematical process shown to enhance the contrast related to myelin content[27,28,88]. The resulting T1-weighted/T2-weighted ratio of the voxels within the cortical ribbon is mapped onto the mid-thickness surface (the latter of which was created by averaging the white and pial surfaces) to reduce partial volume effects. This overall process produces T1-weighted/T2-weighted ratio volumetric as well as surface-based 'intracortical myelin maps' in both the subject's native as well as standardized space.

In order to extract brain regions' intracortical myelin content, we set each subject's surface-based 'intracortical myelin map' and the cortical parcellation of interest mapped into the same space (standardized fsaverage_LR32k space) as inputs for the *wb_command -cifti-parcellate* and *-cifti-convert -to-text* commands. The latter generated a text file for each atlas containing each brain region's ID and its corresponding average T1-weighted/T2-weighted ratio signal intensity; the signal intensity was used as a proxy of that region's intracortical myelin content.

**Excitation-inhibition balance.** The balance between synaptic excitation and inhibition at the neuronal or neuronal circuit level broadly refers to the relative amounts of excitatory and inhibitory synaptic inputs at that level, at any given time scale[80]. It is typically expressed as the ratio of excitatory to inhibitory inputs. This EI-ratio is under tight neuromodulatory control and is critical for circuit function and stability; deviations outside a narrow range have been reported to be pathogenic[46,80,94,95]. Previous work using models of neuronal networks has indicated that changes in EI-ratio are captured by the spectral properties of the recorded electrophysiological signal activity and particularly by the exponent of its 1/f spectral power law, an index that is mathematically related to the signal time series' Hurst exponent[46,96]. This relationship between the Hurst exponent and EI-ratio was also validated in (i) simulated functional BOLD signal data, as well as (ii) resting-state functional BOLD data obtained from mice while chemogenetically manipulating the excitability of their pyramidal neurons; according to that relationship, a heightened EI-ratio would then be reflected as a decrease in the Hurst exponent of the functional signal[46].

Because changes in the Hurst exponent of resting-state fMRI time series can be interpreted as a shift in synaptic EI-ratio[46], we computed the Hurst exponent of each brain region's pre-processed resting-state signal time series and used it as a proxy of the overall EI-ratio within that region. The methodological approach used to perform this computation is described in detail elsewhere[46]. In brief, for each atlas and for each subject, each brain region's pre-processed resting-state signal time series were modeled as multivariate fractionally integrated processes, and the corresponding Hurst exponent was estimated via the univariate maximum likelihood method and a discrete wavelet transform[46,97].

**Voxel-based approach.** We used a voxel-based approach to analyze the Penn sample. For each subject, we first assessed their structural connectivity, then their functional connectivity, and lastly their SFC. In this case—and in contrast to the atlas-based approach—the different types of connectivity were investigated at the cortical voxel level where each subject's cortical voxel was defined as a separate brain region.

**Structural connectivity.** The Penn subjects' high-resolution DSI scans were first pre-processed using *QSIPrep* (https://qsiprep.readthedocs.io/en/latest/; version 0.8.0)[98]. Initial motion correction was performed using only the $b = 0$ images; an unbiased *b0* template was constructed over three iterations of affine registrations. Then, the SHORELine method was used to estimate head motion in $b > 0$ images[99]. This procedure entails leaving out each $b > 0$ image and reconstructing the others using *3dSHORE*[100]; the signal for the left-out image served as the registration target. A total of two iterations were run using an affine transform. Model-generated images were transformed into alignment with each $b > 0$ image. Both slice-wise and whole-brain Quality Control measures (cross correlation and $R^2$) were calculated. A deformation field to correct for susceptibility distortions was estimated based on *fMRIPrep*'s fieldmap-less approach. The deformation field resulted after co-registering the *b0* reference to the same-subject T1-weighted-reference with its intensity inverted[101]. Registration was performed with *antsRegistration* (ANTs 2.3.1), and the process was regularized by constraining deformation to be nonzero only along the phase-encoding direction and modulated with an average field map template. Based on the estimated susceptibility distortion, an unwarped *b0* reference was calculated for a more accurate co-registration with the anatomical reference. Each subject's DSI time series were resampled to AC-PC orientation, generating a pre-processed DSI run in AC-PC space (output space: T1-weighted image; output resolution: 1.8 mm isotropic). After the diffusion scans were pre-processed, *QSIPrep* was used to estimate the diffusion orientation distribution functions

(dODF) at each voxel, using generalized *q*-sampling imaging with a mean diffusion distance of 1.25 mm.

After pre-processing and reconstructing the diffusion scans, we invoked the *MITTENS* Python library (https://github.com/mattcieslak/MITTENS) to perform analytic tractography on the reconstructed DSI data[102]. In contrast to deterministic and probabilistic tractography, this recently established tractography approach calculates connection probabilities between different brain regions without relying on extensive simulations. Given each voxel's dODF and a set of a priori anatomical/geometric constraints, analytic tractography can be used to derive closed-form solutions to the tracking problem, directly computing voxel-to-voxel transition probabilities[102]. First, we calculated inter-voxel fiber transition probabilities by using the reference *bO* image generated by the pre-processing stage and the diffusion dODF output by the reconstruction stage of *QSIPrep* as inputs (maximum turning angle = 35 degrees; step size in voxel units = $\frac{\sqrt{3}}{2}$). This process outputs volumetric (nifti) files for each neighbor direction. We then constructed directed graphs for each voxel, where edges were formed to each of that voxel's 26 spatial neighbors and weighted by the negative logarithm of transition probability from one voxel to another, all while taking into account the dODF of both the source and destination voxels (*double-ODF* method). After supplying *MITTENS* with a cortical mask where each cortical voxel was designated with a different index, the likelihood that a cortical voxel was connected to any other cortical voxel was calculated as the geometric mean of the product of the transition probabilities along the shortest path between the two voxels; the shortest path between voxels was found using Dijkstra's algorithm[103]. A structural connectivity matrix was thus generated for each subject, where each row and column corresponded to a different cortical voxel; each entry was set equal to the likelihood that that voxel was connected to any other cortical voxel. Because each subject had a different number of cortical voxels, the resulting structural connectivity matrices ranged in size between 60,744 × 60,744 and 83,680 × 83,680, depending on the subject. Given the substantial number of brain regions (and their potential interactions) considered, we thresholded our structural connectivity matrices in order to mitigate the presence of spurious connections that could have potentially biased our results[104]. We specifically applied density-based thresholding where we kept 70% of the strongest edges in the connectome and set all others to zero.

**Functional connectivity.** The Penn group's resting-state fMRI scans were pre-processed using the *CONN* (https://web.conn-toolbox.org/home; version 20.b) toolbox[105,106]. We specifically ran *CONN*'s "default pre-processing pipeline for volume-based analyses (direct normalization to MNI-space)." Each subject's functional scans were first co-registered and resampled to a reference image (set as the first scan of the first session). A slice-timing correction procedure then followed, correcting for any potential temporal misalignment that may have occurred during the sequential acquisition of the fMRI data; acquisitions with a framewise displacement above 0.9 mm or global BOLD signal changes above 5 standard deviations were flagged as potential outlier scans. The structural scans were segmented into gray matter, white matter, and cerebrospinal fluid tissue classes using SPM (version 12), and both structural and functional scans were subsequently normalized into MNI space (180 × 216 × 180 mm³ bounding box; functional scans set to 2 mm isotropic; structural scans set to 1 mm isotropic). Lastly, the functional images were smoothed using an 8 mm full-width half-maximum Gaussian kernel, in order to increase the BOLD signal-to-noise ratio[106]. *CONN*'s default denoising pipeline was then applied, which used linear regression of potential confounders identified in the BOLD signal and temporal high-pass filtering. Potential confounding effects that were regressed out of the BOLD signal time series included noise components from white matter and cerebrospinal areas, estimated subject motion

parameters (i.e., 3 rotation and 3 translation parameters, and their 6 associated first-order derivatives), identified outlier scans from the pre-processing step, as well as session-related effects (such as constant and linear BOLD signal trends). Temporal frequencies below 0.008 Hz were also removed from the BOLD signal in order to mitigate the effects of low-frequency drifts. Denoising outputs were manually inspected to ensure approximately centered distributions of the resulting functional connectivity data.

We then registered the pre-processed, denoised functional image from MNI into the subject's *bO* reference image created by *MITTENS* in our structural connectivity analyses. The same cortical mask as the one supplied to *MITTENS* was then overlaid onto the registered functional image, in order to extract the BOLD signal time series corresponding to each cortical voxel, for each subject. Similarly to the atlas-based approach, a 'static' functional connectivity matrix was computed by calculating the Pearson's correlation between the signal time series of any two given cortical voxels. Voxel-based 'temporally-contiguous' functional connectomes (20 per subject) were also generated as described in the atlas-based approach.

**Exclusion criteria.** After the structural and functional scans had been pre-processed and denoised, we manually examined the Quality Control files exported by *MITTENS* and *CONN*, to assess the quality of the data. Given that structural and functional connectivities were being assessed at the voxel-level, we chose to apply particularly conservative quality control criteria when deciding which subjects to include in our analyses: subjects with at least one "bad" slice found (i.e., slices that significantly differed in intensity patterns from the slices acquired before and after)[98] in the pre-processed diffusion images (*n* = 2) or resting-state functional scans with mean framewise displacement exceeding 0.2 mm (*n* = 3) were excluded from the analysis[107,108]. Using these criteria, we included 9 (88.9% female; mean age = 22.8 ± 2.7 years; age range = 19–27 years) of the total 14 subjects scanned with both diffusion spectrum and resting-state functional imaging.

**Structure-function coupling.** The SFC of each subject's cortical voxel was defined as the Spearman's correlation coefficient between the matrix row corresponding to that voxel in the thresholded structural connectome and the matrix row corresponding to that voxel in the 'static' functional connectome, after excluding the self-connection and any other entries where either the regional structural or functional connectivity was equal to zero. Using this definition of SFC and the same approach as the one described in the atlas-based analyses, we also computed each cortical voxel's moment-to-moment (temporal) SFC variance across 20 contiguous non-overlapping time windows.

**Intracortical myelination.** Intracortical myelination in the Penn sample was assessed using the previously validated HCP Pipeline (https://github.com/Washington-University/HCPpipelines/wiki/Installation-and-Usage-Instructions#running-the-hcp-pipelines-on-example-data; version 4.3.0). Specifically, the scripts in the three HCP pipelines (*PreFreeSurfer*, *FreeSurfer*, and *PostFreeSurfer*) were run to generate the T1-weighted/T2-weighted 'myelin maps' for each subject as a proxy for their intracortical myelin content. The individual steps performed by each pipeline have been described above in our atlas-based approach. For each subject, the resulting T1-weighted/T2-weighted ratio volumetric file was then registered from MNI into the subject's *bO* reference image (obtained from *MITTENS*); the signal intensity at each cortical voxel was then extracted using the same cortical mask as the one used in our voxel-based structural and functional connectivity analyses. In order to exclude voxels that might potentially represent non-brain tissue or voxels with aberrantly high or low signal intensity, we only kept values within one standard deviation away from the mean signal intensity of the non-zero intensity voxels within the cortical ribbon mask[28].

**Excitation-inhibition balance.** The same pipeline used to estimate the EI-ratio in our atlas-based approach—in the form of the functional signal time series' Hurst exponent—was also used here. Specifically, each cortical voxel's pre-processed resting-state signal time series were modeled as multivariate fractionally integrated processes, and the corresponding Hurst exponent was estimated via the univariate maximum likelihood method and a discrete wavelet transform[46,97].

**Cortical hierarchies**

To examine the regional distribution of the variables of interest across the cortex, we assigned each brain region (as defined in the 'Atlas-based approach' and 'Voxel-based approach' sections above) an index representing its putative placement along the broader cognitive representational and cyto-architectural hierarchy. For that purpose, we utilized four complementary cortical annotations: (1) 7 resting-state systems (visual, somatomotor, dorsal attention, ventral attention, limbic, fronto-parietal, and default mode) estimated by intrinsic functional connectivity[37] (resting-state systems; coarse metric), (2) the principal gradient of cortical organization derived by the decomposition of connectivity data and intrinsic geometry of the cortex[38] (principal functional gradient; continuous metric), (3) 5 cyto-architectonic classes/types (agranular, frontal, parietal, polar, and granular) derived from cellular morphological properties[39] (von Economo/Koskinas-inspired cyto-architectonic classes; coarse metric), and (4) the "Big-Brain" cortical gradient derived by modeling the similarity of cortical columns' microstructural profiles[40,41] (BigBrain gradient of microstructure profile covariance; continuous metric). The first two annotations spatially group brain regions along the unimodal (sensory)–transmodal (association) hierarchy based on their functional connectivity profiles; the latter two assign brain regions into the same cortical class/type based on their cellular morphological profiles. The membership of each brain region into each of the four mentioned cortical annotations was assigned in the following way:

**Atlas-based approach**

**Schaefer ROI → Resting-state systems.** The assignment of each Schaefer (400 parcels) ROI into its corresponding resting-state system (1–7 systems) was provided as part of the Schaefer atlases download (https://github.com/ThomasYeoLab/CBIG/tree/master/stable_projects/brain_parcellation/Schaefer2018_LocalGlobal)[42].

**Schaefer ROI → Principal functional gradient.** The Schaefer (400 parcels) atlas (filename: Schaefer2018_400Parcels_7Networks_order_FSLMNI152_2mm.nii.gz) and the principal functional gradient (filename: volume.grad_1.MNI2mm.nii.gz) were provided by their respective downloads (mentioned above), in the same space. We then extracted the principal gradient scalars corresponding to all voxels within a given Schaefer ROI and computed their mean, which was then set as the average principal gradient scalar of that Schaefer ROI.

**Schaefer ROI → Cyto-architectonic classes.** The CSV files containing vertices' assignments to the Schaefer 400 parcels and von Economo/Koskinas-inspired cyto-architectonic parcellations (sampled on the standardized *Conte69* surface template) were downloaded from the *ENIGMA* toolbox[109] (https://enigma-toolbox.readthedocs.io/en/latest/index.html). Using these files, we extracted the cyto-architectonic assignments corresponding to all vertices within each Schaefer ROI and computed their mode; the corresponding mode was set as the cyto-architectonic assignment of that Schaefer ROI.

**Schaefer ROI → "BigBrain" gradient.** The BigBrain gradient scalar corresponding to each Schaefer ROI was calculated as previously described[40,41] and provided as part of the *ENIGMA* toolbox as a CSV file.

**HCP multi-modal ROI → Resting-state networks.** The HCP multi-modal atlas in cifti file format was first mapped to the resting-state functional systems in the same format and grayordinates space (RSN-networks.32k_fs_LR.dlabel.nii; https://balsa.wustl.edu/study/show/WG33), using the *Connectome Workbench* toolbox (*wb_command -cifti-create-dense-from-template*). We then extracted the resting-state assignments corresponding to all grayordinates within a given HCP multi-modal ROI and computed their mode; the corresponding mode was set as the resting-state system assignment of that HCP multi-modal ROI[110].

**HCP multi-modal ROI → Principal functional gradient.** We used the principal functional gradient in the same grayordinate space as the HCP multi-modal atlas (cifti file format: hcp.gradients.dscalar.nii; https://github.com/neuroanatomyAndConnectivity/gradient_analysis). We then extracted the principal gradient scalars corresponding to all grayordinates within a given HCP multi-modal ROI and computed their mean, which was then set as the average principal gradient scalar of that HCP multi-modal ROI.

**HCP multi-modal ROI → Cyto-architectonic classes.** The CSV files containing vertices' assignments to the HCP multi-modal and the von Economo/Koskinas-inspired cyto-architectonic parcellations (sampled on the standardized *Conte69* surface template) were downloaded from the *ENIGMA* toolbox[109] (https://enigma-toolbox.readthedocs.io/en/latest/index.html). Using these files, we extracted the cyto-architectonic assignments corresponding to all vertices within each HCP multi-modal ROI and computed their mode; the corresponding mode was set as the cyto-architectonic assignment of that HCP multi-modal ROI.

**HCP multi-modal ROI → "BigBrain" gradient.** The BigBrain gradient scalar corresponding to each HCP multi-modal ROI was calculated as previously described[40,41] and publicly provided as part of the *ENIGMA* toolbox as a CSV file.

**Voxel-based approach**

**Cortical voxels → Resting-state systems.** Using the guidelines provided in the resting-state systems' online documentation[37] (https://github.com/ThomasYeoLab/CBIG/tree/master/stable_projects/brain_parcellation/Yeo2011_fcMRI_clustering/1000subjects_reference/Yeo_JNeurophysiol11_SplitLabels/project_to_individual), we registered the resting-state systems from the standardized *fsaverage* space into each subject's volumetric space (in *FreeSurfer* terminology: their orig.mgz space). Afterwards, we registered those systems into each subject's reference *b0* space (generated by *MITTENS*) using the *antsApplyTransforms* command (*ANTs* 2.3.1) with the "MultiLabel" interpolation flag. The resulting atlas was dilated three times to ensure that all cortical voxels were assigned a resting-state system affiliation. Lastly, using this dilated atlas and the same cortical ribbon mask as the one mentioned in our voxel-based structural and functional connectivity analyses, we extracted each cortical voxel's resting-state system affiliation.

**Cortical voxels → Principal functional gradient.** We first registered the principal functional gradient (volume.grad_1.MNI2mm.nii.gz; https://github.com/neuroanatomyAndConnectivity/gradient_analysis) from MNI into each subject's reference *b0* space (generated by *MITTENS*) using the *antsApplyTransforms* command (*ANTs* 2.3.1). The registered gradient was dilated once to ensure that all cortical voxels were assigned a gradient scalar. Lastly, using this dilated atlas and the same cortical ribbon mask as the one mentioned in our voxel-based structural and functional connectivity analyses, we extracted each cortical voxel's corresponding principal gradient scalar.

**Cortical voxels → Cyto-architectonic classes.** The von Economo/Koskinas-inspired cyto-architectonic atlas was downloaded in MNI ICBM 2009a Nonlinear Symmetric stereotaxic space (http://www.dutchconnectomelab.nl)[111]. We then registered this atlas into each subject's reference *b0* space (generated by *MITTENS*) using the *antsApplyTransforms* command (*ANTs* 2.3.1) with the "MultiLabel" interpolation flag. The resulting atlas was dilated three times in order to ensure that all cortical voxels were assigned a cyto-architectonic cortical type affiliation. Lastly, using this dilated atlas and the same cortical ribbon mask as the one mentioned in our voxel-based structural and functional connectivity analyses, we extracted each cortical voxel's cyto-architectonic cortical type affiliation.

## Statistical analyses

Statistical analyses were performed using the *SPSS* statistical software (version 28: IBM Corp.), *MATLAB* (version R2021a: The MathWorks, Inc.), and *Python* (version 3.7).

**Atlas-based approach: dataset.** For our 'Atlas-based approach' analyses delineated above, we generated and analyzed two datasets: one wherein each row corresponded to each Schaefer ROI (for a total of 400 rows) and another wherein each row corresponded to each HCP multi-modal ROI (for a total of 360 rows). Each dataset's column corresponded to the variable of interest (e.g., SFC, temporal SFC variance, intracortical myelin content, and Hurst exponent) averaged across subjects (unless otherwise specified above).

**Voxel-based approach: dataset.** For our voxel-based approach, we generated one dataset for each Penn subject that passed our quality control assessments (for a total of 9 datasets), as each subject had a different number of cortical voxels. Within each dataset, each row corresponded to each cortical voxel of that subject, and each column reflected the variable of interest corresponding to that subject's cortical voxel.

Each statistical analysis described below was applied separately to each one of these 9 datasets. Correlation and regression coefficients corresponding to each dataset were then averaged and a mean value was reported. In order to combine the *p*-values generated for each analysis pertaining to each dataset (subject) into one representative combined *p*-value, we applied Fisher's method of meta-analysis[112]. This method entailed calculating first the following test statistic *T* with a $\chi^2$-distribution and 18 degrees of freedom (=number of datasets x 2):

$$T = -2\sum_{i=1}^{n} \ln(p_i) \qquad (1)$$

where *ln* is the natural logarithm and $p_i$ the *p*-value corresponding to dataset *i*. The combined *p*-value (referred to as $p_{fisher}$ in the manuscript) is then calculated as follows:

$$p_{fisher} = 1 - \chi^2_{cdf}(T; v = 2n) \qquad (2)$$

where $\chi^2_{cdf}$ is the cumulative distribution function (cdf) for a $\chi^2$-distribution with $v$ degrees of freedom (here, $n = 9$)[113,114].

**ANOVA tests.** One-way analysis of variance (ANOVA) tests were used to statistically compare the overall differences in SFC and temporal SFC variance across the 7 resting-state systems and 5 cyto-architectonic classes, described in the 'Structure-Function Coupling Variations along the Cortical Hierarchy' section of the Results. The ANOVA tests were followed by *post-hoc* correction for multiple comparisons (Tamhane's T2−equal variances not assumed) analyses to examine the statistical differences between all possible pairs among the resting-state systems and all possible pairs among the cyto-architectonic classes.

**Bivariate analyses and spatial permutation tests.** Comparisons between the four variables of interest: SFC, temporal SFC variance, intracortical myelin content, and the Hurst exponent, were carried out in the form of previously established spatial permutation tests (threshold for significance: $p < 0.05$)[115,116]. In contrast to bivariate correlations such as Spearman's or Pearson's, spatial permutation tests take into account the potential spatial autocorrelation that might exist between variables and neighboring brain regions as well as hemispheric symmetry, by generating a set of appropriate spatial autocorrelation-preserving null models for each hemisphere. Specifically, the empirical two-tailed Spearman's correlation between any two spatial maps (i.e., two variables) is compared to a distribution of null Spearman's correlations, generated by projecting one of the spatial maps into a sphere, randomly rotating that sphere, and then projecting the rotated spherical map back onto the brain surface[115,116]. In our study, this 'spin test' was repeated 10,000 times to generate 10,000 null correlations, for each comparison. The empirical Spearman's correlation coefficient (*r*) and the *p*-value derived by comparing the empirical with the null correlations (referred to in the manuscript as $p_{spin}$) were reported for each bivariate comparison described in our atlas-based analyses. In the voxel-based analyses, we reported a mean *r*, its [min, max] range across the 9 subjects, and the combination of all subjects' $p_{spin}$ values into one combined $p_{fisher}$ value, as described in the "Voxel-based approach: Dataset" section above.

Furthermore, we also tested the assumption of homoscedasticity in our analyses (i.e., the assumption that the variance of the residuals in the regression model is constant as the independent variable changes) using the Breusch-Pagan test: we (i) first fit the regression model using our empirical dependent and independent variables, (ii) calculated the square of the unstandardized residuals of the model, and (iii) then fit a new regression model using the squared residuals as the new dependent variable. The *p*-value between the squared residuals and the independent variable was then calculated for each subject; these *p*-values were then combined into one $p_{fisher}$ value, as described in the "Voxel-based approach: Dataset" section above.

**Multiple linear regression analyses and non-parametric bootstrapping.** Multiple linear regression models were used to examine the statistical relationship between two variables, after adjusting for the effects of other pertinent variables. SFC and temporal SFC variance were designated as the dependent variables, whereas intracortical myelin content and the Hurst exponent were designated as the independent variables. To account for the presence of an interaction effect between these two independent variables, we included their interaction effect in the multiple linear regression models described in the 'Biological Correlates of Structure-Function Coupling: Whole-brain perspective' section, as an additional independent variable. We specifically (i) created centered versions of the two variables using the following formula:

$$X_{i,centered} = X_i - average(\mathbf{X}) \qquad (3)$$

where $X_i$ is the variable of interest (here, intracortical myelin content or Hurst exponent) corresponding to brain region *i*, and **X** is a vector corresponding to the variable's values across all brain regions. We then (ii) computed an 'interaction effect' set equal to the product of the two centered variables, and (iii) used the centered intracortical myelin content, centered Hurst exponent, their interaction effect, and the two gradient assignments of cortical hierarchy placement (the principal functional gradient and "BigBrain" gradient assignments) as the independent variables. The latter two variables were included to ensure that any potential relationships were not driven by a similar co-variation of the given variables across the same cortical hierarchy. Centered variables of intracortical myelin content and the Hurst

exponent were used to mitigate any potential multicollinearities among the independent variables. Although there were no significant multicollinearities present in the atlas-based analyses when the interaction term was included (i.e., Variance Inflation Factor [*VIF*] < 2 – see the end of this sub-section)[117], there were significant multi-collinearities introduced in the voxel-based analyses (*VIF* values of intracortical myelin content and interaction effect > 46, across subjects). Hence, we did not include the interaction term between intracortical myelin content and the Hurst exponent as an additional independent variable in our voxel-based multiple regression models.

Moreover, to address the non-linear relationship and significant heteroscedasticity between the temporal SFC variance and the Hurst exponent in our voxel-based analyses, we incorporated a non-linear term (square of the Hurst exponent) as an additional independent variable in the 'Biological Correlates of Structure-Function Coupling: Whole-brain perspective' and 'Biological Correlates of Structure-Function Coupling: Regional perspective' sections.

Standardized $\beta$ ($\beta_{stand}$) coefficients and *p*-values were computed for each independent variable within each multiple linear regression model (ordinary least squares regression), using non-parametric bootstrapping. This process entailed (i) fitting the original empirical data into the multiple regression model and calculating the $\beta_{stand}$ coefficients, (ii) sampling the original empirical data with replacement, and (iii) re-fitting the multiple regression model on this newly sampled dataset and extracting the resulting $\beta_{stand}$ coefficients. We repeated steps (ii)-(iii) 10,000 times to generate robust confidence intervals for the $\beta_{stand}$ coefficients and the corresponding 'bootstrapped' *p*-values. For each multiple regression model mentioned in the atlas-based analyses of our Results section, we reported the empirically derived $\beta_{stand}$ coefficient for each independent variable, the corresponding 95% confidence interval as calculated by the non-parametric boot-strapping approach (and referred to in the text as 95% *BCI*), and the resulting *p*-value. Similarly, for our voxel-based analyses, we reported the mean empirically-derived $\beta_{stand}$ coefficient for each independent variable across subjects, its [min, max] range across the 9 subjects, and the combination of all subjects' *p*-values into one combined $p_{fisher}$ value, as described in the "Voxel-based approach: Dataset" section above. Overall, applying non-parametric bootstrapping into our regression models allowed us to robustly examine how variable the $\beta_{stand}$ coefficients were in each model, without making any assumptions about the distribution of the data.

Lastly, to ensure that there were no collinearities among our variables within the multiple regression models, we also reported the *VIF* within each analysis; a threshold of *VIF* > 5 was used to indicate significant collinearity[117].

**Mediation model.** The mediation analysis reported in our atlas-based analyses in the Results section: 'Biological Correlates of Structure-Function Coupling: Whole-brain perspective' was performed using the *PROCESS* (v3.4) statistical macro for *SPSS*[118]. Intracortical myelin content was designated as the independent variable, the Hurst exponent of the functional signal time series as the mediator, and the temporal SFC variance as the dependent variable. The Hurst exponent (as a proxy of EI-ratio) was chosen as the mediator in this model–rather than the independent variable–as it fluctuates on a moment-to-moment basis[80]. The hypothesized mediation effect was tested using bootstrapping (10,000 samples); a *BCI* that did not include zero indicated a significant mediation effect.

**Citation diversity statement**
Recent work in several fields of science has identified a bias in citation practices such that papers from women and other minority scholars are under-cited relative to the number of such papers in the field[119–123]. We obtained the predicted gender of the first and last author of each reference by using databases that store the probability of a first name

being carried by a woman[123]. By this measure (and excluding self-citations to the first and last authors of our current paper), our references contain 12% woman(first)/woman(last), 9.3% man/woman, 21.4% woman/man, and 57.3% man/man. This method is limited in that (a) names, pronouns, and social media profiles used to construct the databases may not, in every case, be indicative of gender identity and (b) it cannot account for intersex, non-binary, or transgender people. We look forward to future work that could help us better understand how to support equitable practices in science.

**Reporting summary**
Further information on research design is available in the Nature Portfolio Reporting Summary linked to this article.

## Data availability
The Human Connectome Project dataset used in this study is publicly available at https://db.humanconnectome.org/. The University of Pennsylvania sample dataset analyzed as well as the scripts generated for the purposes of this study are available from the corresponding authors upon request. Source data are provided with this paper.

## Code availability
A copy of the custom-made scripts used to perform the main atlas- and voxel-based analyses described in this manuscript–as well as instructions on how to run them–can be found in: https://github.com/pfotiad/SFC_Nature_Comm_2023.git.

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

## Acknowledgements

Data were provided in part by the Human Connectome Project, WU-Minn Consortium (Principal Investigators: David Van Essen and Kamil Ugurbil; 1U54MH091657) funded by the 16 National Institutes of Health (NIH) Institutes and Centers that support the NIH Blueprint for Neuroscience Research; and by the McDonnell Center for Systems Neuroscience at Washington University. This work was supported by NIH grants T32-EB020087 (P.F.), R37MH125829 (T.D.S.), R01MH113550 (T.D.S. & D.S.B.), RF1MH116920 (T.D.S. & D.S.B.), R21MH106799 (T.D.S. & D.S.B.), R01MH112847 (R.T.S. & T.D.S.), R01MH112847 (R.T.S.), R01MH123550 (R.T.S.), and R01NS112274 (R.T.S.), the Research Start-up Fund of the University of Science and Technology of China (X.H.), the National Natural Scientific Foundation of China (No. 82271491; X.H.), the Swartz Foundation (D.S.B.), and the John D. and Catherine T. MacArthur Foundation (D.S.B.).

## Author contributions

Conceptualization: P.F. and D.S.B. Methodology: P.F., M.C., X.H., L.C., M.O., T.D.S., R.T.S., D.S.B. Data Analyses, Writing – Original Draft: P.F. Writing – Review & Editing: M.C., X.H., L.C., M.O., T.D.S., R.T.S., D.S.B. Supervision: D.S.B.

## Competing interests

The authors declare no competing interests.
