## [Peer Review File · Nature Communications]

Myelination and excitation-inhibition balance synergistically shape structure-function coupling across the human cortexReviewer #1 (Remarks to the Author):

In this study, the authors investigated the probably biological factors underlying the spatial layout of the structure-function coupling (SFC) in the brain, which was derived according to the correlation between the structural connectivity pattern and the functional connectivity pattern in the same brain location. They found that intracortical myelination and excitation inhibition (EI) balance might contribute to explaining both the static SFC and the dynamic SFC in both the whole-brain and the regional perspectives. Multimodal MRI data and other public datasets describing the brain hierarchical organization were adopted here. Although the topic is interesting, I still have several major/minor concerns below.

1. It seems that the HCP data were used for the atlas-based approach only, and, the Penn samples were used for the voxel-based approach only. In this way, we are not completely assured about the consistency between the atlas-based approach and the voxel-based approach. Why not just test the analyses in the same way in both datasets (i.e., both having the atlas-based and the voxel-based analyses)?

2. Regarding the temporal SFC, the moment-to-moment variance was calculated to represent the temporal dynamic of the SFC. However, the four runs of the HCP resting-state fMRI were concatenated to calculate the above variance. Would it be possible that the difference between runs could contribute to the derived variance? In addition, is there any reason to select '20 continuous non-overlapping time windows' for calculating the variance?

3. Given that the temporal SFC was based on the non-overlapping time windows, should the 'Hurst exponent and EI-ratio' be also observed in each time window? Otherwise, the Hurst exponent of the whole time series may not directly account for the variance of the temporal SFC as the EI-ratio may also have temporal change. Again, whether the concatenated data may affect the calculation of the Hurst exponent is not totally clear.

4. In the multiple linear regression analyses, the principal functional gradient was included as an independent variable. However, it is not clear why the "BigBrain" gradient was excluded, especially given that it also showed a significant correlation with SFC.

5. It is not clear whether the mediation model should be used to explain the relationship between temporal SFC variance, intracortical myelination, and the Hurst exponent. An interaction effect between intracortical myelination and the Hurst exponent seems more feasible, especially given the myelination and E-I ratio relationship in Fig. 6.

6. The authors should include a schematic diagram to show the analytic pipeline and the related datasets more clearly, given that multiple analytic-level and brain measures were included here.

Reviewer #2 (Remarks to the Author):

In this nice paper the authors quantify structure-function coupling (SFC) across the cortex and ask:

- 1) Whether SFC correlates with macro-scale functional features and microscale cytoarchitectonic features
- 2) Whether SFC correlates with intra-cortical myelination and E-I ratio
- 3) Whether the correlations with myelination and E/I differ across different cytoarchitectonic regions

I thought the paper very clear and well-written and the analyses seem technically sound and robust to various processing options. Indeed it is a strength of the paper that they used multiple datasets and also systematically considered different parcellations as well as voxel-based analyses – showing that all options produce convergent results.

Personally, I felt the paper was a little 'thin' in terms of results and in-depth discussion and I would have liked to see one or two additional substantial results, especially if published in this journal. However this is a very subjective assessment, so it should only be considered if it is brought up by multiple reviewers.

My more specific questions and comments are as follows:

1. In the introduction (line ~88) the authors describe how “the weaker SFC in higher-level association cortices is thought to foster the emergence of a wide range of functional responses untethered from the underlying anatomical backbone, in turn supporting flexible cognition”. I wonder how this is reconciled with the previous statement that higher working memory is associated with increased SFC in the default-mode network and decreased SFC in unimodal sensory regions (line 71). Could the authors clarify?

2. In Fig 2 C and D I was curious whether the effects are driven by some outlier regions?

3. I was surprised by the definition of E/I as the Hurst exponent of the functional timeseries. How closely does this correspond to other estimates of E/I balance, for instance based on gene expression?

4. I was a little unsure about the motivation behind focusing on myelination and on E/I balance. The introduction says: “Recent evidence suggests that the differential expression of neuronal circuit properties—including intracortical myelination and synaptic excitation or inhibition—could serve as such biological substrates. Histological and neuroimaging studies show that high SFC areas in primary sensory and motor cortex are heavily myelinated, whereas lower SFC areas in association cortex are less myelinated. Following a similar spatial pattern, synaptic excitation increases from unimodal sensory to transmodal association cortex, tracking a concomitant increase in dendritic complexity and spine count. Further, immunostaining investigations tracking the differential expression of inhibitory neuron subtypes, evince a unimodal-transmodal gradient of dynamic inhibitory control. Put together, the ratio between excitatory and inhibitory receptor densities (EI-ratio) appears to increase along the sensory-association hierarchy. It remains unknown, however, whether the differential expression of intracortical myelination and EI-ratio formally mediate the observed differences in macroscale SFC across the cortex.” This seems problematic in two ways:

First, it seems that the motivation for focusing on these two biological features is that their spatial variation appears (by eye) correlated with that of SFC – the current manuscript then quantifies this correlation. Is this the case, or is there a more hypothesis-driven reason to focus on these features?

Secondly, it is well-known that many biological features correlate with the broad gradient sometimes called the S-A hierarchy. Having tested only these two features I am not sure how convincing the suggested causal relationship to SFC is. Can the authors consider other candidate mechanisms (preferable others also known to vary across the S-A axis and show some kind of specificity)?

5. I thought the discussion was well-written, but a large fraction of it focused on recapping results – it would have been more interesting to find more additional context or interpretations in this section. For example, there is a large literature on SFC using different definitions for how it is quantified. It would be useful to have a bit more context on how these different approaches (and different results) are related. Secondly it would be useful to have a bit more discussion on the mechanisms that might underpin the results presented.

RESPONSE TO REVIEWER COMMENTS

Reviewers' comments are shown in black type; authors' comments are shown in red type.

Reviewer #1:

In this study, the authors investigated the probably biological factors underlying the spatial layout of the structure-function coupling (SFC) in the brain, which was derived according to the correlation between the structural connectivity pattern and the functional connectivity pattern in the same brain location. They found that intracortical myelination and excitation inhibition (EI) balance might contribute to explaining both the static SFC and the dynamic SFC in both the whole-brain and the regional perspectives. Multimodal MRI data and other public datasets describing the brain hierarchical organization were adopted here. Although the topic is interesting, I still have several major/minor concerns below.

We would like to thank the reviewer for their interest, and have aimed to address their concerns as thoroughly as possible.

1. It seems that the HCP data were used for the atlas-based approach only, and, the Penn samples were used for the voxel-based approach only. In this way, we are not completely assured about the consistency between the atlas-based approach and the voxel-based approach. Why not just test the analyses in the same way in both datasets (i.e., both having the atlas-based and the voxel-based analyses)?

Response:

Our motivation for utilizing our (originally) three different processing pipelines to analyze the HCP sample and Penn samples was to examine whether our results converge no matter what processing pipeline is used; and in that sense show that our results are not confounded by varying methodological aspects, but rather that they can be replicated among different populations and across different processing pipelines.

Besides this more general motivation, we also chose to not originally apply the atlas-based methodology on our Penn sample considering its rather small sample size ($n = 14$). We do, however, agree with the reviewer that it would be beneficial to the reader to also see the atlas-based methodology applied on this sample.

We have now re-analyzed the Penn sample using the same atlas-based processing pipeline as the one applied on the HCP sample. Our results remained consistent, and have been added into our supplemental materials (section name: **Supplemental**

Analysis 3, p. 9-11). Moreover, we added the following sections into our main manuscript pointing to this additional analysis:

Results section: (p. 5):

“To evaluate the reproducibility of our findings, we repeated the above analyses (i) using a different widely-used cortical parcellation (HCP multi-modal parcellation; Supplemental Material: Supplemental Analysis 1), (ii) using a complementary definition of functional signal time series (see **Methods: Processing Pipelines: Functional Connectivity**; Supplemental Material: Supplemental Analysis 2), and (iii) on the complementary Penn sample to establish generalizability across different subject samples (Supplemental Material: Supplemental Analysis 3). We observed consistent results across all supplemental analyses.”

Results section: (p. 7):

“To assess reproducibility and robustness across different processing pipelines and subject samples, we repeated all aforementioned analyses (i) using the HCP multi-modal cortical parcellation, (ii) using a complementary definition of functional signal time series (see **Methods: Processing Pipelines: Functional Connectivity**), and (iii) across the Penn sample, and observed consistent results (**Supplemental Analyses 1, 2, 3**).”

2. Regarding the temporal SFC, the moment-to-moment variance was calculated to represent the temporal dynamic of the SFC. However, the four runs of the HCP resting-state fMRI were concatenated to calculate the above variance. Would it be possible that the difference between runs could contribute to the derived variance? In addition, is there any reason to select '20 continuous non-overlapping time windows' for calculating the variance?

Response:

The reviewer raises an excellent point. To ensure that our results are not confounded—in general—by the concatenation of the four fMRI runs obtained by the HCP dataset, we repeated all of the analyses reported in the manuscript with one main difference: instead of concatenating the four runs across time (1200 volumes x 4 runs), we averaged the demeaned and normalized pre-processed time series corresponding to the four runs into one average run (1200 volumes). We then used this averaged time series as the functional signal to re-compute the SFC, temporal SFC variance, and Hurst exponent corresponding to the HCP sample, using both cortical parcellations described in the manuscript: the Schaefer parcellation and the HCP multi-modal parcellation. We replaced all results in our main manuscript corresponding to the Schaefer parcellation, added the results corresponding to the HCP multi-modal parcellation as a new section in the Supplemental Materials, entitled “Supplemental Analysis 1,” and updated the “Functional Connectivity” sub-section of our “Methods:

Processing Pipelines: Atlas-based approach” section accordingly, to indicate that the functional runs were averaged, rather than concatenated. We also moved all the results that were previously reported in the manuscript, using the concatenated signal time series, into the Supplemental Material, in the section entitled “Supplemental Analysis 2” (parts A and B). Overall, the results across all of these complementary analyses remained qualitatively the same.

Concerning the reviewer’s second point regarding our choice to use 20 temporal windows, we have added the following clarifying segment into our “**Supplemental Materials: Methodological Considerations and Study Limitations**” section (p. 11):

“Moreover, our choice to measure the temporal SFC variance by splitting the duration of the functional scan into 20 continuous temporal windows served the purpose of generating time windows that captured a few minutes of functional activity. Given the differences in functional scan duration and repetition time across the two participant groups (Human Connectome Project [HCP] and Penn samples) and across our processing pipelines, each of the 20 time windows corresponded to ~40 seconds of functional activity in the atlas-based analyses (HCP sample) described in the main manuscript and Supplemental Analysis 1, ~3 minutes of functional activity in the atlas-based analyses (HCP sample) described in Supplemental Analyses 2A and 2B, and 1 minute of functional activity in the atlas- and voxel-based analyses derived from analyzing the Penn sample (Supplemental Analysis 3 and main manuscript). Even though the duration of each time window could be expected to influence the corresponding temporal SFC variance, our results remained consistent across all analyses utilizing varying window durations. Future studies, however, should further vary the number of time windows (with or without temporal overlap) and examine whether—and to what extent—this choice impacts the resulting SFC variability over time.”

3. Given that the temporal SFC was based on the non-overlapping time windows, should the 'Hurst exponent and EI-ratio' be also observed in each time window? Otherwise, the Hurst exponent of the whole time series may not directly account for the variance of the temporal SFC as the EI-ratio may also have temporal change. Again, whether the concatenated data may affect the calculation of the Hurst exponent is not totally clear.

Response:

We chose to non-invasively estimate the EI-ratio using the Hurst exponent of the entire fMRI signal time-series (rather than the Hurst exponent of the time-series corresponding to each one of the individual windows) for the following reasons:

- 1) As a metric, the Hurst exponent assesses the autocorrelation across the functional signal (i.e., the cross-correlation of the fMRI signal with itself at different points in time, estimated across a range of time lags of the signal).³

Thus, the more time points we have available, the more accurately the Hurst exponent can capture the ‘long-memory’ properties of the signal. In our particular datasets, the fMRI signal time-series consisted of (1) 1,200 volumes for the atlas-based analyses reported in the main manuscript (HCP sample), (2) 2,400 volumes for the Penn sample, and (3) 4,800 volumes for the analyses reported in Supplemental Analyses 2A and 2B. Given that the discrete wavelet transform applied to calculate the Hurst exponent of these time-series uses volumes in powers of 2,^{4,5} the first $2^{10} = 1,024$ volumes were utilized in case (1), the first $2^{11} = 2,048$ volumes were utilized in case (2), and the first $2^{12} = 4,096$ volumes were utilized in case (3) to compute the Hurst exponent of the corresponding time series. If we had, however, calculated the Hurst exponent on each of the 20 non-overlapping time windows used in the temporal SFC calculations, we would end up relying upon a significantly smaller number of volumes ($2^5 = 32$ volumes in case (1), $2^6 = 64$ volumes in case (2), and $2^7 = 128$ volumes in case (3)), rendering the calculation of the Hurst exponent of that segment of the time-series potentially unreliable.

- 2) One of the questions of interest in the manuscript was whether the extent of temporal fluctuations of SFC (i.e., temporal SFC variance) was dependent upon the overall levels of (relative) excitation or inhibition characterizing a brain region. If we were to compute the Hurst exponent (and thus EI-ratio) at each temporal window and then computed its variance across time—similarly to what we did with SFC—we wouldn’t necessarily be able to directly address this question of interest, but rather whether SFC and EI-ratio co-fluctuate over a short period of time (i.e., the duration of the fMRI scan).

We do, however, agree with the reviewer that it would still be informative to examine whether SFC and EI-ratio co-fluctuate over short periods of time (i.e., the duration of the fMRI scan), across our two samples. For that purpose, we computed the SFC and Hurst exponent (as a proxy of EI-ratio) of each brain region, at each time window, across our two samples, and examined whether they displayed a significant correlation across time. It should be noted that given our response to the reviewer’s second question, we are now calculating the SFC and Hurst exponent on the averaged (rather than concatenated) functional time series of the HCP sample.

Overall, SFC and the Hurst exponent did not co-fluctuate throughout the duration of the fMRI scan in our atlas-based analyses (Schaefer parcellation: average Spearman’s ρ across brain regions: 0.04; $p_{fisher} (FDR\ corrected)=1$, and HCP multi-modal parcellation: average Spearman’s ρ across brain regions: -0.02; $p_{fisher} (FDR\ corrected)=1$). In our voxel-based analyses, however, there was a significant positive association between SFC and the Hurst exponent (average Spearman’s ρ across subjects and across brain regions: 0.03; p range across subjects: [0.009, 0.08]; $p_{fisher} (FDR\ corrected)<0.001$) across the different time windows.

We are now reporting these results in our main manuscript:

Results section (p. 6-7):

“Notably, there was no significant association between SFC and the Hurst exponent across the different temporal windows (average Spearman’s ρ across brain regions: 0.04; $p_{fisher} (FDR\ corrected)=1$), indicating that SFC and EI-ratio do not co-fluctuate over short periods of time (i.e., the duration of the fMRI scan), when examined on the macroscale level.”

Results section (p. 8):

“Lastly—and in contrast to the atlas-based analyses—there was a significantly positive association between SFC and the Hurst exponent across the different temporal windows (average Spearman’s ρ across subjects and across brain regions: 0.03; ρ range across subjects: [0.01, 0.08]; $p_{fisher} (FDR\ corrected)<0.001$), indicating that SFC and EI-ratio co-fluctuate over short periods of time (i.e., the duration of the fMRI scan), when examined under the finer spatial scale defined by our voxel-based analysis.”

And in our Supplemental Material: Supplemental Analysis 1 (p. 3):

“There was no significant association, however, between SFC and the Hurst exponent across temporal windows (average Spearman’s ρ across brain regions: -0.02; $p_{fisher} (FDR\ corrected)=1$), indicating that SFC and EI-ratio do not co-fluctuate over short periods of time (i.e., the duration of the fMRI scan), when examined on the macroscale level.”

4. In the multiple linear regression analyses, the principal functional gradient was included as an independent variable. However, it is not clear why the “BigBrain” gradient was excluded, especially given that it also showed a significant correlation with SFC.

Response:

To address the reviewer’s point, we have now repeated our multiple linear regression analyses including the “BigBrain” gradient as an additional independent variable, and have updated the corresponding β_{stand} , 95% non-parametric bootstrap confidence intervals, and p -values in the Results section of our main manuscript (p. 7), as well as in Supplemental Analyses 1, 2, and 3 (p. 3, 5, 6, 8, 10).

5. It is not clear whether the mediation model should be used to explain the relationship between temporal SFC variance, intracortical myelination, and the Hurst exponent. An interaction effect between intracortical myelination and the Hurst exponent seems more feasible, especially given the myelination and E-I ratio relationship in Fig. 6.

Response:

We absolutely agree with the reviewer that, given our conclusions in Figure 6 (now Figure 7), it would be statistically rigorous to include an interaction effect between intracortical myelination and the Hurst exponent in our regression models. We have now added such an interaction effect by: 1) creating two new variables, each set equal to the biological marker of interest, centered (i.e., centered variable = raw variable – average(raw variable across all brain regions)), 2) computing an interaction effect, set equal to the product of the two centered variables of interest (intracortical myelin content and the Hurst exponent), and 3) repeating our multiple linear regression models described in the atlas-based analyses in the sections entitled: “Biological Correlates of Structure-Function Coupling: Whole-brain perspective” found in our main manuscript as well as in Supplemental Analyses 1, 2, and 3, with the centered intracortical myelin content, centered Hurst exponent, interaction effect between the two variables, and the two cortical hierarchy scalars (principal functional gradient and BigBrain gradient), as the independent variables. After conducting these analyses, we updated the corresponding β_{stand} , 95% non-parametric bootstrap confidence intervals, and p -values (Main manuscript: p. 7; Supplemental Material: p. 3, 5, 6, 8, 10).

Furthermore, we found and reported a significant interaction effect between intracortical myelination and the Hurst exponent taking place in the models predicting the temporal SFC variance, in the atlas-based analyses reported in the main manuscript (p. 7) as well as Supplemental Analyses 2A and 2B (Supplemental Material: p. 6 and 8). We chose not to add an interaction effect between myelination and the Hurst exponent in our voxel-based analyses, given how the resulting VIF of the independent variables ‘intracortical myelination’ and ‘interaction effect’ were extremely high ($VIF > 46$ in all models), indicating severe collinearity between the two.

In addition to our Results section, we have also updated our **Methods section (p. 25)** to reflect that such interaction effects were considered in the atlas-based but not in the voxel-based analyses:

“To account for the presence of an interaction effect between these two independent variables, we included their interaction effect in the multiple linear regression models described in the ‘Biological Correlates of Structure-Function Coupling: Whole-brain perspective’ section, as an additional independent variable. We specifically (i) created centered versions of the two variables using the following formula:

$$X_{i,centered} = X_i - average(\mathbf{X})$$

where X_i is the variable of interest (here, intracortical myelin content or Hurst exponent) corresponding to brain region i , and \mathbf{X} is a vector corresponding to the variable’s values across all brain regions. We then (ii) computed an ‘interaction effect’ set equal to the product of the two centered variables, and (iii) used the centered intracortical myelin content, centered Hurst exponent, their interaction effect, and the two gradient assignments of cortical hierarchy placement (the principal functional gradient and “BigBrain” gradient assignments) as the independent variables. The latter two variables were included to ensure that any potential relationships were not driven by a similar co-

variation of the given variables across the same cortical hierarchy. Centered variables of intracortical myelin content and the Hurst exponent were used to mitigate any potential multicollinearities among the independent variables. Although there were no significant multicollinearities present in the atlas-based analyses when the interaction term was included (i.e., Variance Inflation Factor [VIF] < 2 — see the end of this sub-section), there were significant multicollinearities introduced in the voxel-based analyses (VIF values of intracortical myelin content and interaction effect > 46 , across subjects). Hence, we did not include the interaction term between intracortical myelin content and the Hurst exponent as an additional independent variable in our voxel-based multiple regression models.”

Lastly, we chose to still keep the mediation models as an additional analysis that could further complement and elucidate the interplay between intracortical myelin and Hurst exponent in predicting temporal SFC variance.

6. The authors should include a schematic diagram to show the analytic pipeline and the related datasets more clearly, given that multiple analytic-level and brain measures were included here.

Response:

The reviewer raises an important point that would make our manuscript significantly clearer. We have now added a new figure into our manuscript (**Figure 1**) schematically illustrating the datasets used, the corresponding analytic pipelines used to process these datasets, and the location of the results of each analysis.

Reviewer #2:

In this nice paper the authors quantify structure-function coupling (SFC) across the cortex and ask:

- 1) Whether SFC correlates with macro-scale functional features and microscale cytoarchitectonic features
- 2) Whether SFC correlates with intra-cortical myelination and E-I ratio
- 3) Whether the correlations with myelination and E/I differ across different cytoarchitectonic regions

I thought the paper very clear and well-written and the analyses seem technically sound and robust to various processing options. Indeed it is a strength of the paper that they used multiple datasets and also systematically considered different parcellations as well as voxel-based analyses – showing that all options produce convergent results.

We would like to thank the reviewer for their kind words.

Personally, I felt the paper was a little ‘thin’ in terms of results and in-depth discussion

and I would have liked to see one or two additional substantial results, especially if published in this journal. However this is a very subjective assessment, so it should only be considered if it is brought up by multiple reviewers.

To address the reviewer's concerns, we have now added multiple new analyses, results, and discussion points into our manuscript and the supplemental material, as can be seen from our responses to Reviewer #1 and to the questions below.

My more specific questions and comments are as follows:

1. In the introduction (line ~88) the authors describe how “the weaker SFC in higher-level association cortices is thought to foster the emergence of a wide range of functional responses untethered from the underlying anatomical backbone, in turn supporting flexible cognition”. I wonder how this is reconciled with the previous statement that higher working memory is associated with increased SFC in the default-mode network and decreased SFC in unimodal sensory regions (line 71). Could the authors clarify?

Response:

We would like to thank the reviewer for raising this excellent point. We have now modified our statement to more accurately incorporate previous findings in the field, and specifically replaced:

Introduction (p. 3):

“The weaker SFC in higher-level association cortices is thought to foster the emergence of a wide range of functional responses untethered from the underlying anatomical backbone, in turn supporting flexible cognition.”

with:

“The presence of a dynamic SFC landscape along the sensory-association hierarchy is thought to foster the emergence of a wide range of functional responses untethered from the underlying anatomical backbone, in turn supporting flexible cognition.”

2. In Fig 2 C and D I was curious whether the effects are driven by some outlier regions?

Response:

To address the reviewer's important point, we identified the brain regions displaying outlier 'temporal SFC variance' values in Figures 2C and D (now Figures 3C and D), and repeated the correlation analyses reported in these figures after excluding outlier values. In this context, an outlier brain region was defined as one that exhibited a temporal SFC variance at least three standard deviations away from the mean.

When applied to the dataset used to generate Figures 2C and D (now Figures 3C and D), this process identified 7 brain regions as outliers. We excluded these 7 brain regions, and repeated the analyses reported in these figures. Consistent with the results reported in the main manuscript, both correlations remained significant (temporal SFC variance vs. principal functional gradient: Spearman's $r=0.42$; $p_{spin}<0.001$, and temporal SFC variance vs. "BigBrain" gradient: Spearman's $r=0.40$; $p_{spin}=0.005$).

To ensure that outlier regions were not driving the association between temporal SFC variance and the two gradient scalars when using the Glasser parcellation (**Supplemental Analysis 1**), we repeated the aforementioned process for the results shown in Supplementary Figures 2C and D. In this case there were 7 brain regions that were identified as outliers, with respect to their temporal SFC variance. As before, both correlations remained significant (temporal SFC variance vs. principal functional gradient: Spearman's $r=0.40$; $p_{spin}=0.007$, and temporal SFC variance vs. "BigBrain" gradient: Spearman's $r=0.46$; $p_{spin}=0.015$) after removal of the outliers—with variance explained remaining the same as when including the outliers in the analysis.

Similarly, we repeated the same analyses for **Supplemental Analysis 2A** (i.e., Supplemental Figures 5C and D), **Supplemental Analysis 2B** (i.e., Supplemental Figures 8C and D), and **Supplemental Analysis 3** (i.e., Supplemental Figures 11C and D); all results were consistent with those obtained when outliers were included in the analyses.

Collectively, these results indicate that outlier brain regions (i.e., brain regions with outlier temporal SFC variance) did not significantly impact the association between temporal SFC variance and the gradient scalars portrayed in Figures 2C and D (now Figures 3C and D) or the corresponding supplemental figures.

We have now incorporated all these new results into our revised manuscript:

Main Manuscript: Results (p. 5):

"To ensure that the correlations observed between a brain region's temporal SFC variance and its location in the sensory-association hierarchy (as shown in Figures 3C and 3D) were not confounded by the presence of any outlier regions, we repeated the aforementioned analyses after excluding the outlier brain regions. An outlier brain region was defined as one that exhibited a temporal SFC variance at least three standard deviations away from the mean ($n=7$). Consistent with our results when the outliers were included, temporal SFC variance was significantly correlated with both the principal functional gradient ($r=0.42$; $p_{spin}<0.001$) and the BigBrain gradient ($r=0.40$; $p_{spin}=0.005$)." View text

Supplemental Material: Supplemental Analysis 1 (p. 2-3):

“As with the Schaefer atlas, in order to ensure that the correlations observed between a brain region’s temporal SFC variance and its location in the sensory-association hierarchy (as shown in Supplementary Figures 2C and 2D) were not confounded by the presence of any outlier regions, we repeated the aforementioned analyses after excluding the outlier brain regions. An outlier brain region was defined as one that exhibited a temporal SFC variance at least three standard deviations away from the mean ($n=7$). Both correlations remained significant (temporal SFC variance vs. principal functional gradient: $r=0.40$; $p_{spin}=0.007$, and temporal SFC variance vs. BigBrain gradient: $r=0.46$; $p_{spin}=0.015$)—same as when including the outliers in the analysis.”

Supplemental Material: Supplemental Analysis 2A (p. 5):

“To ensure that the correlations observed between a brain region’s temporal SFC variance and its location in the sensory-association hierarchy (as shown in Supplementary Figures 5C and 5D) were not confounded by the presence of any outlier regions, we repeated the aforementioned analyses after excluding the outlier brain regions. As before, an outlier brain region was defined as one that exhibited a temporal SFC variance at least three standard deviations away from the mean ($n=8$). Both correlations remained qualitatively the same: the association between temporal SFC variance and principal functional gradient remained positive, albeit non-significant ($r=0.17$; $p_{spin}=0.108$), whereas the association between temporal SFC variance and the BigBrain gradient remained significant ($r=0.29$; $p_{spin}=0.040$)—same as when the outliers were included.”

Supplemental Material: Supplemental Analysis 2B (p. 7):

“As with the Schaefer atlas, in order to ensure that the correlations observed between a brain region’s temporal SFC variance and its location in the sensory-association hierarchy (as shown in Supplementary Figures 8C and 8D) were not confounded by the presence of any outlier regions, we repeated the aforementioned analyses. After identifying the outlier regions ($n=9$), we excluded them and repeated the analyses reported in these figures. Both correlations remained significant (temporal SFC variance vs. principal functional gradient: $r=0.29$; $p_{spin}=0.049$, and temporal SFC variance vs. BigBrain gradient: $r=0.44$; $p_{spin}=0.027$)—same as when including the outliers in the analysis.”

Supplemental Material: Supplemental Analysis 3 (p. 9-10):

“Similar to Supplemental Analyses 1 and 2, we wanted to ensure that the correlations observed between a brain region’s temporal SFC variance and its location in the sensory-association hierarchy (as shown in Supplementary Figures 11C and 11D) were not confounded by the presence of any outlier regions. After identifying the outlier regions ($n=10$), we excluded them and repeated the analyses reported in these figures. Both correlations remained qualitatively the same as when the outliers were included: the association between temporal SFC variance and principal functional gradient

remained significant ($r=0.22$; $p_{spin}=0.047$), whereas the association between temporal SFC variance and the BigBrain gradient remained insignificant ($r=0.21$; $p_{spin}=0.084$).”

3. I was surprised by the definition of E/I as the Hurst exponent of the functional timeseries. How closely does this correspond to other estimates of E/I balance, for instance based on gene expression?

Response:

Indeed, our choice to non-invasively assess excitation/inhibition (E/I) balance in our samples using the functional blood oxygen-level dependent (BOLD) timeseries’ Hurst exponent was motivated by recent work combining *in silico* and *in vivo* chemogenetic manipulations of neuronal excitation and inhibition to study sex-related heterogeneity of E/I (im)balance in autism.^{4,6} In this work, the authors first showed that in a biologically-representative computational neuronal model, the Hurst exponent of the simulated local field potentials (LFP) and BOLD signals could accurately capture changes in the underlying synaptic E/I conductance ratio.^{4,6} Then, the authors empirically tested these predictions *in vivo*. They specifically measured the resting-state functional BOLD signals in mice while chemogenetically manipulating the excitability of pyramidal neurons in their prefrontal cortex. Agreeing with the *in silico* simulations, the *in vivo* experiments demonstrated that the Hurst exponent of the BOLD signal timeseries could accurately track changes in the experimentally manipulated E/I ratio.⁴ Lastly, the authors used the Hurst exponent of the BOLD signal timeseries of individuals with autism as a proxy of their E/I (im)balance, to investigate sex-specific differences in its expression.⁴ Therefore, as shown by this work, the Hurst exponent of the functional timeseries closely corresponds to other computational and experimental estimates of E/I balance, lending support to its merit as a non-invasive proxy of E/I ratio.

To address the reviewer’s critical point, and to further indicate the agreement between this definition of E/I balance (i.e., the Hurst exponent) and other computational and experimental estimates of E/I balance, we have now replaced the following segment in our **‘Methods: Excitation-Inhibition Balance’ section (p. 17):**

“This relationship between the Hurst exponent and EI-ratio was also validated in simulated functional BOLD signal data; according to that relationship, a heightened EI-ratio would then be reflected as a decrease in the Hurst exponent of the functional signal.”

with:

“This relationship between the Hurst exponent and EI-ratio was also validated in (i) simulated BOLD signal data, as well as (ii) resting-state functional BOLD data obtained from mice while chemogenetically manipulating the excitability of their pyramidal neurons; according to that relationship, a heightened EI-ratio would then be reflected as a decrease in the Hurst exponent of the functional signal.”

4. I was a little unsure about the motivation behind focusing on myelination and on E/I balance. The introduction says: “Recent evidence suggests that the differential expression of neuronal circuit properties—including intracortical myelination and synaptic excitation or inhibition—could serve as such biological substrates. Histological and neuroimaging studies show that high SFC areas in primary sensory and motor cortex are heavily myelinated, whereas lower SFC areas in association cortex are less myelinated. Following a similar spatial pattern, synaptic excitation increases from unimodal sensory to transmodal association cortex, tracking a concomitant increase in dendritic complexity and spine count. Further, immunostaining investigations tracking the differential expression of inhibitory neuron subtypes, evince a unimodal-transmodal gradient of dynamic inhibitory control. Put together, the ratio between excitatory and inhibitory receptor densities (EI-ratio) appears to increase along the sensory-association hierarchy. It remains unknown, however, whether the differential expression of intracortical myelination and EI-ratio formally mediate the observed differences in macroscale SFC across the cortex.” This seems problematic in two ways:

First, it seems that the motivation for focusing on these two biological features is that their spatial variation appears (by eye) correlated with that of SFC – the current manuscript then quantifies this correlation. Is this the case, or is there a more hypothesis-driven reason to focus on these features?

Response:

Indeed, our choice of biological features of interest was hypothesis-driven; we would like to thank the reviewer for giving us the opportunity to further highlight this point.

Our motivation behind choosing intracortical myelination as one of the two biological features of interest was four-fold:

1. Previous work has pointed towards a systematic relationship across the sensory-association hierarchy between myelination and functional connectivity.^{7,8}
2. Changes of structure-function coupling across normative development have led authors of previous work to propose that myelination might be a driving factor—given that it also dramatically changes across the same time period.^{9,10}
3. As described in our Introduction section (and mentioned above by the reviewer), the spatial patterns of intracortical myelination across the cortex appear to match those of structure-function coupling.
4. A relationship between structure-function coupling and intracortical myelination would make biological sense, given how increased myelination acts as an insulator of signal propagation and has been proposed to inhibit synaptic plasticity, thus potentially constraining the emergence of functional signals that deviate from structural paths (as mentioned in our ‘Discussion section: Biological Substrates of Structure-Function Coupling’ in p. 11).^{11,12}

Our motivation behind choosing EI-ratio as the second biological feature of interest was also four-fold:

1. Recent work has pointed towards different neurotransmitters as potentially regulating the loss of structure-function coupling observed in patients with Parkinson's disease.¹³ Given that neurotransmitters typically adjust the EI-ratio of neuronal clusters, we thought that the latter biological feature would be involved in shaping structure-function coupling across the cortex.
2. A neuronal cluster's EI-ratio has been directly linked to its functional output, as described in more detail in our 'Discussion section: Biological Substrates of Structure-Function Coupling' in p. 11. Therefore, we assumed that it would also play an important role in determining to what extent a brain region's functional connectivity deviates from its underlying structural connectivity.
3. As described in our Introduction section (and mentioned above by the reviewer), the spatial patterns of EI-ratio expression across the cortex appear to broadly match those of structure-function coupling.
4. Lastly, structure-function coupling and its temporal variance change across multiple time scales (i.e., fast and short time scales) whereas intracortical myelination typically changes across shorter time scales. We therefore chose to additionally include a biological feature that changes across fast time scales to properly capture structure-function coupling. Given that EI-ratio is such a feature—and given our previous considerations—we chose to include it as another biological feature.

To address the reviewer's comment and make our motivation behind choosing intracortical myelination and EI-ratio as our two biological features of interest more explicit in the manuscript, we have now included the following paragraphs in our Introduction section:

Introduction (p. 3-4):

“Moreover, regional heterogeneities in intracortical myelination have been linked to differences in functional connectivity patterns across the cortical mantle; brain regions with similar intracortical myelin profiles typically display stronger functional connectivity to each other.^{7,8} This correspondence is particularly high within unimodal brain regions; transmodal regions such as the posteromedial cortex, the anterior insular cortex, and the superior portions of the inferior parietal lobule, instead, display a lower correspondence between intracortical myelination and functional connectivity, even after correcting for inter-regional proximity.⁸ Lastly, the relationship between structural and functional connectivity drastically changes throughout normative development—a critical period of enhanced neuroplasticity and myelination—which could point towards intracortical myelination's potential involvement as one of its mediators.^{9,10}

Besides intracortical myelination, neuromodulation has also been implicated as a potential driving factor determining to what extent the brain's functional expression is tethered to the underlying anatomical connectivity. Following a similar spatial pattern as

intracortical myelination, synaptic excitation increases from unimodal sensory to transmodal association cortex, tracking a concomitant increase in dendritic complexity and spine count.¹⁴ Further, immunostaining investigations tracking the differential expression of inhibitory neuron subtypes, evince a unimodal-transmodal gradient of dynamic inhibitory control.^{14,15} Put together, the ratio between excitatory and inhibitory receptor densities (EI-ratio) appears to increase along the sensory-association hierarchy.¹⁶ What is more, recent work looking into the differences in SFC between patients with Parkinson's disease and healthy controls identified an increased association between the expression of various neurotransmitter receptor genes and disease-related structure-function decoupling.¹³ Thus, given that such neuromodulatory systems typically alter the balance between the excitation and inhibition of their targeted neuronal circuits, we postulated that EI-ratio would also play an important role in shaping the healthy human brain's SFC.

The aforementioned observations collectively motivated our hypothesis that the differential expression of intracortical myelination and EI-ratio formally mediate the heterogeneous expression of SFC across the cortex.”

Secondly, it is well-known that many biological features correlate with the broad gradient sometimes called the S-A hierarchy. Having tested only these two features I am not sure how convincing the suggested causal relationship to SFC is. Can the authors consider other candidate mechanisms (preferable others also known to vary across the S-A axis and show some kind of specificity?

Response:

Although absolutely an interesting point to consider, we wanted to focus on a few select biological features and extensively investigate their potential contributions in shaping structure-function coupling. We do acknowledge, however, that other biological features could certainly contribute to the way the brain's structural connectivity shapes its functional expression, and *vice versa*. Certain such examples could include (i) cyto-architectonic properties, such as the underlying neuronal density, neuronal size, and firing behavior (e.g., tonic versus burst firing) patterns found in different brain regions, and (ii) other neuromodulatory properties, such as the contribution of various neurotransmitters and neuropeptides, and their heterogeneous effects on different brain regions. All of these features are heterogeneously expressed across the sensory-association hierarchy, and examining their specific contributions in shaping structure-function coupling could serve as an intriguing future direction.

To incorporate the reviewer's concern into our manuscript, we have now replaced the following segment in our '**Supplemental Materials: Methodological Considerations and Study Limitations**' section (p. 12-13):

“Lastly, even though the goal of this study was to identify the biological substrates that mediate how strongly coupled the functional connectivity is to the structural connectivity,

we do acknowledge that—in addition to intracortical myelination and EI-ratio—there could be a number of other biological markers that could contribute towards this coupling.”

with:

“Lastly, even though the goal of this study was to identify the biological substrates that mediate how strongly coupled the functional connectivity is to the structural connectivity, we do acknowledge that—in addition to intracortical myelination and EI-ratio—there could be a number of other biological markers that could contribute towards this coupling. Specific examples could include (i) cyto-architectonic properties, such as the underlying neuronal density, neuronal size, and firing behavior (e.g., tonic versus burst firing) patterns found in different brain regions, and (ii) other neuromodulatory properties, such as the contribution of various neurotransmitters and neuropeptides, and their heterogeneous effects on different brain regions.”

5. I thought the discussion was well-written, but a large fraction of it focused on recapping results – it would have been more interesting to find more additional context or interpretations in this section. For example, there is a large literature on SFC using different definitions for how it is quantified. It would be useful to have a bit more context on how these different approaches (and different results) are related. Secondly it would be useful to have a bit more discussion on the mechanisms that might underpin the results presented.

Response:

We would like to thank the reviewer for their suggestions.

To expand upon the first point raised on the potential impact of different definitions of SFC, we have added the following segment to our **Discussion section (p. 10-11)**:

“Furthermore, studies using definitions of SFC other than the correlational approach utilized in this work have also found a heterogeneous decoupling between structure and function across the sensory-association hierarchy. Such definitions have quantified SFC by invoking (i) spectral graph theory, where a brain region’s functional brain activity (typically the blood oxygen level-dependent [BOLD] signal) is expressed as a weighted linear combination of the harmonic components of the brain’s structural connectome (i.e., as defined by its eigendecomposition); structure-function decoupling can then be assessed as the ratio between the higher spatial frequency ‘decoupled’ and lower spatial frequency ‘coupled’ portions of the spectrum,^{17,18} and (ii) linear regression modeling approaches, where a brain region’s SFC is assessed by how well its empirically-defined functional connectivity can be predicted by linear models incorporating markers of structural organization, such as Euclidean distance, shortest path length, and communicability, obtained from the structural connectome.^{19,20} Such

complementary approaches can be particularly informative in deciphering the spatial and topological attributes of the structural connectome most relevant in mediating SFC.”

Moreover, to expand upon the second point raised by the reviewer on additional discussion on mechanisms underpinning our results, we have added the following segment to our manuscript:

Discussion section (p. 11):

“In turn, the heavy myelination observed in some brain regions, such as the primary sensory and motor cortices, could support these regions’ functional specialization.”

Discussion section (p. 11):

“The enhanced affinity for neuroplasticity within lightly myelinated transmodal regions could thus foster the emergence of flexible functional dynamics characteristic of adaptive behavior and learning.”

Discussion section (p. 13):

“Collectively, the extent to which the spontaneous activity of a brain region is tethered to the underlying white matter projections is evidently shaped by the regional intracortical myelin content and EI-ratio. Intracortical myelination and neuromodulation, however, are highly multi-faceted properties, each representing the concerted sum of other biological properties. Myelination patterns, for instance, rely upon glial-neuronal interactions^{21,22} as well as genetic and environmental influences,^{23,24} and have been shown to extend well into the third decade of life.²⁴ Moreover, neuronal excitation and inhibition patterns are mediated by the release of excitatory (e.g., glutamate) or inhibitory (i.e., gamma-aminobutyric acid) neurotransmitters, and modulated by the activity of major regulatory systems in the central nervous system, such as the dopaminergic, noradrenergic, serotonergic, and cholinergic systems, at any given time. Therefore, it would be critical to examine in future studies how each of these individual facets of neurobiology sculpts the dynamic relationship between structural and functional connectivity in the human brain, at rest or during a task, in health or disease.”

References

1. Power, J. D. *et al.* Methods to detect, characterize, and remove motion artifact in resting state fMRI. *NeuroImage* **84**, 10.1016/j.neuroimage.2013.08.048 (2014).
2. Gu, S. *et al.* Emergence of system roles in normative neurodevelopment. *Proc. Natl. Acad. Sci.* **112**, 13681–13686 (2015).

3. Fornito, A., Zalesky, A. & Bullmore, E. *Fundamentals of Brain Network Analysis*. (Academic Press, 2016).
4. Trakoshis, S. *et al.* Intrinsic excitation-inhibition imbalance affects medial prefrontal cortex differently in autistic men versus women. *eLife* **9**, e55684 (2020).
5. You, W., Achard, S., Stadler, J., Bruekner, B. & Seiffert, U. Fractal analysis of resting state functional connectivity of the brain. *2012 Int. Jt. Conf. Neural Netw.* (2012).
6. Gao, R., Peterson, E. J. & Voytek, B. Inferring synaptic excitation/inhibition balance from field potentials. *NeuroImage* **158**, 70–78 (2017).
7. Hunt, B. A. E. *et al.* Relationships between cortical myeloarchitecture and electrophysiological networks. *Proc. Natl. Acad. Sci.* **113**, 13510–13515 (2016).
8. Huntenburg, J. M. *et al.* A Systematic Relationship Between Functional Connectivity and Intracortical Myelin in the Human Cerebral Cortex. *Cereb. Cortex* **27**, 981–997 (2017).
9. Baum, G. L. *et al.* Development of structure–function coupling in human brain networks during youth. *Proc. Natl. Acad. Sci.* **117**, 771–778 (2020).
10. Vandewouw, M. M., Hunt, B. A. E., Ziolkowski, J. & Taylor, M. J. The developing relations between networks of cortical myelin and neurophysiological connectivity. *NeuroImage* **237**, 118142 (2021).
11. Glasser, M. F., Goyal, M. S., Preuss, T. M., Raichle, M. E. & Van Essen, D. C. Trends and properties of human cerebral cortex: Correlations with cortical myelin content. *NeuroImage* **93**, 165–175 (2014).
12. Sampaio-Baptista, C. & Johansen-Berg, H. White Matter Plasticity in the Adult Brain. *Neuron* **96**, 1239–1251 (2017).

13. Zarkali, A. *et al.* Organisational and neuromodulatory underpinnings of structural-functional connectivity decoupling in patients with Parkinson's disease. *Commun. Biol.* **4**, 86 (2021).
14. Wang, X.-J. Macroscopic gradients of synaptic excitation and inhibition in the neocortex. *Nat. Rev. Neurosci.* **21**, 169–178 (2020).
15. DeFelipe, J., Gonzalez-Albo, M. C., Del Rio, M. R. & Elston, G. N. Distribution and patterns of connectivity of interneurons containing calbindin, calretinin, and parvalbumin in visual areas of the occipital and temporal lobes of the macaque monkey. *J. Comp. Neurol.* **412**, 515–526 (1999).
16. Goulas, A. *et al.* The natural axis of transmitter receptor distribution in the human cerebral cortex. *Proc. Natl. Acad. Sci.* **118**, e2020574118 (2021).
17. Preti, M. G. & Van De Ville, D. Decoupling of brain function from structure reveals regional behavioral specialization in humans. *Nat. Commun.* **10**, 4747 (2019).
18. Griffa, A., Amico, E., Liégeois, R., Van De Ville, D. & Preti, M. G. Brain structure-function coupling provides signatures for task decoding and individual fingerprinting. *NeuroImage* **250**, 118970 (2022).
19. Liu, Z.-Q. *et al.* Time-resolved structure-function coupling in brain networks. *Commun. Biol.* **5**, 1–10 (2022).
20. Vázquez-Rodríguez, B. *et al.* Gradients of structure–function tethering across neocortex. *Proc. Natl. Acad. Sci.* **116**, 21219–21227 (2019).
21. Nave, K.-A. Myelination and support of axonal integrity by glia. *Nature* **468**, 244–252 (2010).

22. Hughes, A. N. Glial Cells Promote Myelin Formation and Elimination. *Front. Cell Dev. Biol.* **9**, 661486 (2021).
23. Schmitt, J. E., Raznahan, A., Liu, S. & Neale, M. C. The Genetics of Cortical Myelination in Young Adults and its Relationships to Cerebral Surface Area, Cortical Thickness, and Intelligence: a Magnetic Resonance Imaging Study of Twins and Families. *NeuroImage* **206**, 116319 (2020).
24. Miller, D. J. *et al.* Prolonged myelination in human neocortical evolution. *Proc. Natl. Acad. Sci.* **109**, 16480–16485 (2012).

Reviewer #1 (Remarks to the Author):

In the revised manuscript, the authors addressed a lot of reviewers' concerns, especially for the first reviewer's questions. However, the fourth question from the second reviewer was not addressed well. That is, why the motivation of this study was focusing attention on "myelination" and "excitation-inhibition ratio", which relates to the core of this article. The authors' response did not provide reasonable evidence to justify the motivation for focusing on these features.

MANUSCRIPT: NCOMMS-22-47095A

SECOND RESPONSE TO REVIEWER COMMENTS

Reviewers' comments are shown in black type; authors' comments are shown in red type.

Reviewer #1 (Remarks to the Author):

In the revised manuscript, the authors addressed a lot of reviewers' concerns, especially for the first reviewer's questions. However, the fourth question from the second reviewer was not addressed well. That is, why the motivation of this study was focusing attention on "myelination" and "excitation-inhibition ratio", which relates to the core of this article. The authors' response did not provide reasonable evidence to justify the motivation for focusing on these features.

Response:

We would like to thank the reviewer for their comments. We do, however, strongly maintain that the reasons listed in our previous response served as critical motivations for choosing intracortical myelination and excitation-inhibition ratio as the two biological variables of interest in this study.